# Disentangling oncogenic amplicons in esophageal adenocarcinoma

Alvin Wei Tian Ng[1,2,3], Dylan Peter McClurg[1], Ben Wesley [4],
Shahriar A. Zamani [1], Emily Black [1], Ahmad Miremadi[5], Olivier Giger[6],
Rogier ten Hoopen[7], Ginny Devonshire [2], Aisling M. Redmond[1], Nicola Grehan[1],
Sriganesh Jammula[1], Adrienn Blasko[1], Xiaodun Li[1], Samuel Aparicio [8,9],
Simon Tavaré [4,10,11], Oesophageal Cancer Clinical and Molecular Stratification
(OCCAMS) Consortium*, Karol Nowicki-Osuch [4] & Rebecca C. Fitzgerald [1] ✉

Esophageal adenocarcinoma is a prominent example of cancer characterized by frequent amplifications in oncogenes. However, the mechanisms leading to amplicons that involve breakage-fusion-bridge cycles and extrachromosomal DNA are poorly understood. Here, we use 710 esophageal adenocarcinoma cases with matched samples and patient-derived organoids to disentangle complex amplicons and their associated mechanisms. Short-read sequencing identifies *ERBB2*, *MYC*, *MDM2,* and *HMGA2* as the most frequent oncogenes amplified in extrachromosomal DNAs. We resolve complex extrachromosomal DNA and breakage-fusion-bridge cycles amplicons by integrating of *de-novo* assemblies and DNA methylation in nine long-read sequenced cases. Complex amplicons shared between precancerous biopsy and late-stage tumor, an enrichment of putative enhancer elements and mobile element insertions are potential drivers of complex amplicons' origin. We find that patient-derived organoids recapitulate extrachromosomal DNA observed in the primary tumors and single-cell DNA sequencing capture extrachromosomal DNA-driven clonal dynamics across passages. Prospectively, long-read and single-cell DNA sequencing technologies can lead to better prediction of clonal evolution in esophageal adenocarcinoma.

The advent of new sequencing technologies means that it is possible to dissect the mechanisms underlying mutations responsible for cancer with increasing precision. Among the classes of mutations, amplifications of oncogenes are critical for the development and progression of many cancers, and they arise through multiple, highly complex mechanisms that have been difficult to unravel. These include segregation errors leading to linear amplifications, breakage-fusion-bridge cycles (BFBs) generating large-scale inversions, and the formation of extrachromosomal DNA (ecDNA)[1–3]. These ecDNA structures are circular, lack telomeres and centromeres, and exhibit stochastic, and hence unequal, segregation in daughter cells. Since the inheritance of ecDNA is not under the strict control seen for chromosomal

[1]Early Cancer Institute, University of Cambridge, Cambridge CB2 0XZ, UK. [2]Cancer Research UK Cambridge Institute, University of Cambridge, Cambridge, UK. [3]Lee Kong Chian School of Medicine, Nanyang Technological University, Singapore, Singapore. [4]Irving Institute for Cancer Dynamics, Columbia University, New York, USA. [5]Department of Histopathology, Cambridge University Hospitals NHS Foundation Trust, Cambridge, UK. [6]Department of Pathology, University of Cambridge, Cambridge CB2 0QQ, UK. [7]Department of Oncology, University of Cambridge, Cambridge CB2 0QQ, UK. [8]Department of Molecular Oncology, British Columbia Cancer Research Centre, Vancouver, British Columbia, Canada. [9]Department of Pathology and Laboratory Medicine, University of British Columbia, Vancouver, British Columbia, Canada. [10]Department of Statistics, Columbia University, New York, USA. [11]Department of Biological Sciences, Columbia University, New York, USA.*A list of authors and their affiliations appears at the end of the paper. ✉e-mail: rcf29@cam.ac.uk

DNA, the emergence of ecDNA can play a vital role in driving intra-tumor heterogeneity by generating cells with a random number of copies of ecDNA[4–6], which can have potent effects in driving the expression of oncogenes[4,5]. More recently, ecDNA amplicons have been shown to coalesce and lead to the higher expression of multiple oncogenes present in these circles[6]. Multiple mechanisms of regulation of the expression of oncogenes present on ecDNA have been reported including enhancer hijacking, trans-activation of enhancers, and interactions with chromosomal enhancers[4–6].

Esophageal adenocarcinoma (EAC) is a poor prognosis cancer that has a predominance of large-scale copy number (CN) alterations including oncogenic amplicons[7–11]. These CN alterations can occur in pre-neoplastic Barrett's esophagus[12], and recently ecDNA events have also been identified in dysplastic Barrett's esophagus in two cohorts, further highlighting the importance of amplicons in the pathogenesis of this disease[13]. Hence, EAC is an ideal cancer type to uncover the plethora of mechanisms leading to oncogenic amplification events in cancer and to study the biological consequences driving tumorigenesis.

Alongside sequencing studies of primary tissues, patient-derived organoid models have been established for EAC. These models recapitulate the genetic lesions found in patients and are representative preclinical models of cancer evolution[14,15]. The organoid models capture the clonal diversity and clonal dynamics of patient tumors[14,16], without contamination by the tumor microenvironment, providing opportunities to study the relationship between gene amplifications and their effects on clonal selection.

In this study, we characterize complex amplicons in EAC to understand their underlying mechanisms in a cohort of 710 primary tumor samples and 24 tumor-derived organoids using short-read WGS sequencing. Additionally, we sequence nine tumors and three organoid samples using Oxford Nanopore long-read sequencing and an organoid model at two-time points using single-cell DNA (scDNA) sequencing (Fig. 1). We apply a combination of technologies to primary tissue and model systems for a detailed inference of mechanisms driving complex amplification, temporal dynamics, and their biological consequences.

## Results

### Analysis of 710 tumors identifies recurrent amplicons and breakpoints in EAC

The cohort consisted of 710 EAC patients undergoing curative treatment. The demographics are representative of this disease with a 5.8:1 male-to-female ratio, an average age of 66.8 years and 43.8% of cases were Stage 3 (Supplementary Table 1, Supplementary Data 1).

Our analysis focused on systematically identifying the genomic regions in EAC that were highly amplified and reconstructing the amplicons using Amplicon Architect[17](AA). We used thresholds (CN > 4.5, region size > 10kbp) that included amplicons present at a lower copy number or diluted due to lower sample cellularity. Next, we determined the frequency of each amplified region in the genome across the cohort and annotated previously identified oncogenic drivers[9–11]. Amplified regions with >3% prevalence across the cohort were selected for detailed classification of the type of amplification events[13] (Fig. 2A).

The majority of amplifications were found to be ecDNA (39%, 241 events) followed by BFBs (29%, 175), complex non-cyclic (20%, 122),

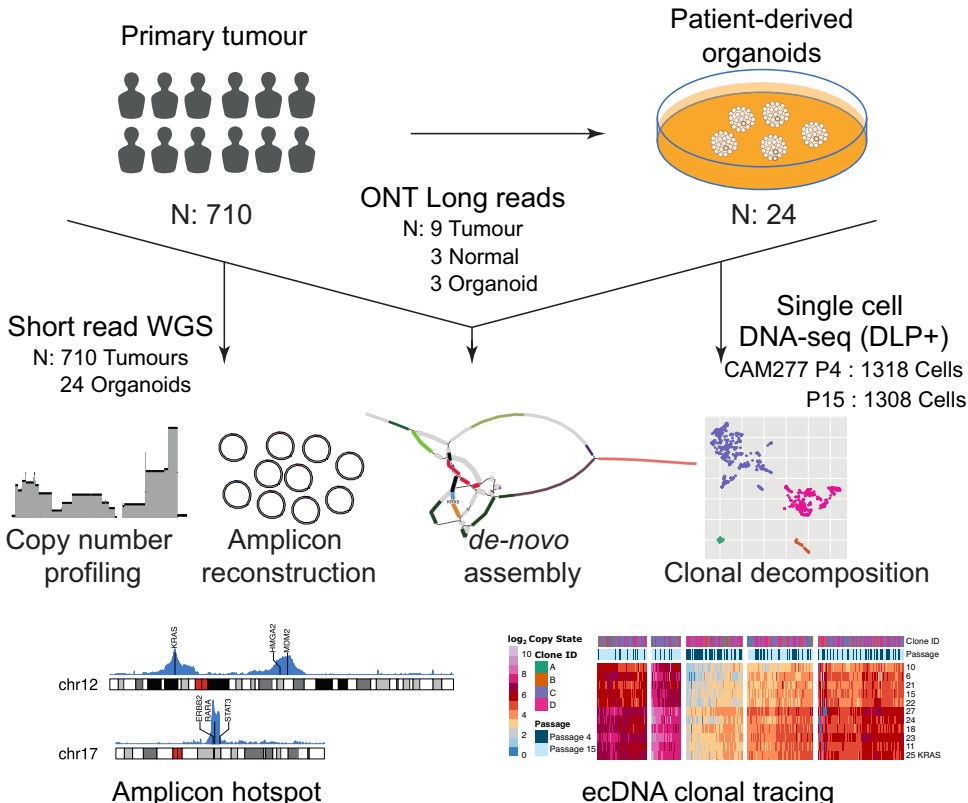

**Fig. 1 | Study design and overview.** Primary tumors (*n* = 710) and patient-derived organoids (*n* = 24) sequenced on Illumina short reads and Oxford Nanopore Technologies (ONT) long-read sequenced tumor, matched normal and organoids (*n* = 9, 3 and 3 respectively) were used in this study. A single-cell DNA sequenced DLP+ library was generated for an organoid at 2 time points. Short-read data was used to profile copy numbers in each sample, reconstruct amplicons using Amplicon Architect and identify amplicon hotspots in the genome. Long reads were used to carry out the *de-novo* assembly of amplicons and used for ecDNA clone tracing in combination with scDNA-seq data. BioRender was used to generate Fig. 1.

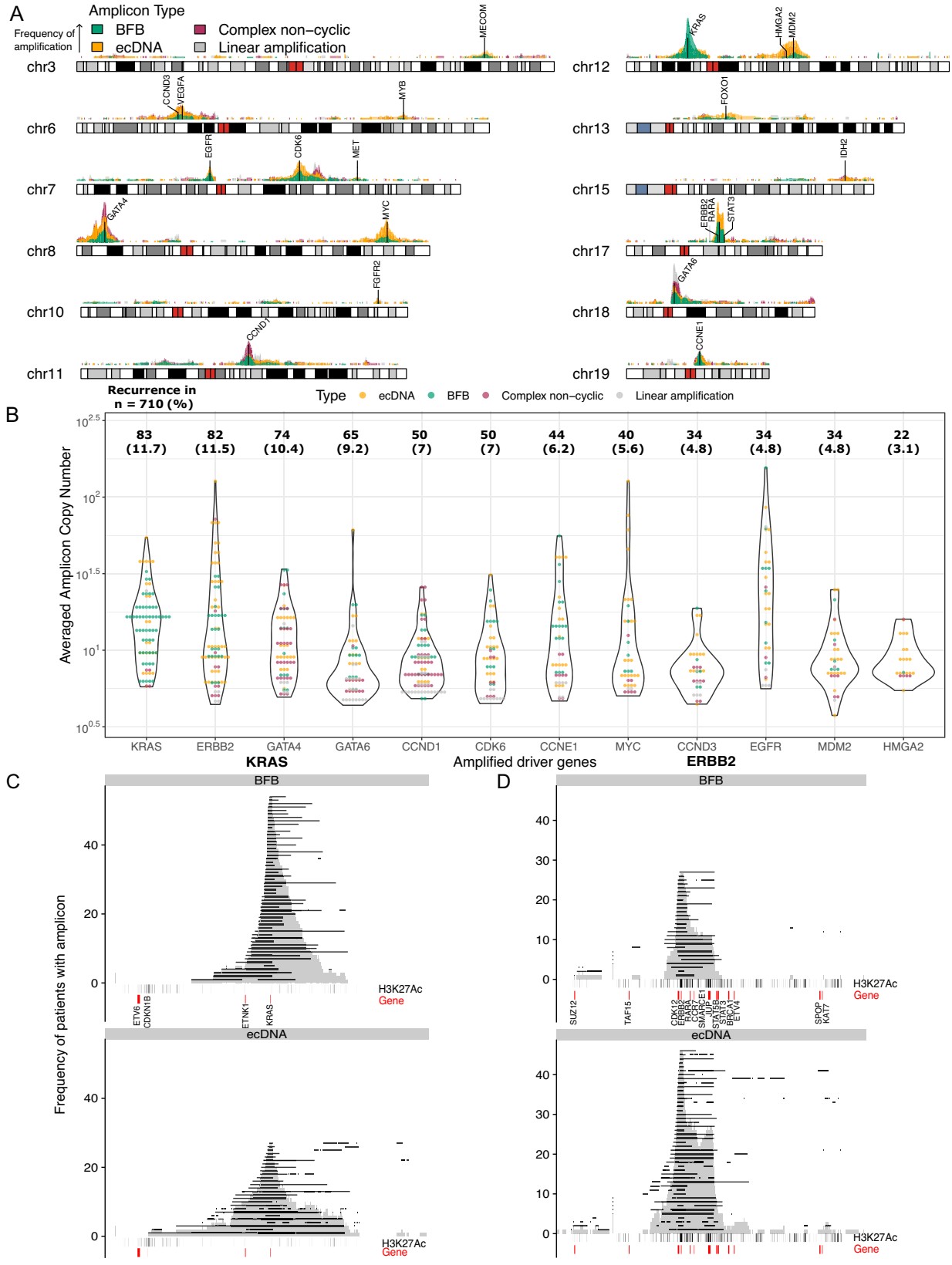

and linear amplification events (12%, 74, Supplementary Table 2). We identified four driver genes (*ERBB2, MYC, MDM2,* and *HMGA2*) that are predominantly altered by ecDNA compared to BFBs (Fig. 2A-B, Supplementary Table 2). Amplicons containing oncogenes *KRAS, ERBB2, MYC, EGFR* and *CCNE1* showed uniform CN distributions with high CN (median CN = 11.3, Interquartile range (IQR) = 7.3-20), compared to

regions containing *GATA6, CCND1, CCND3, MDM2* and *HMGA2* with lower CN (CN = 7.3, IQR = 6.1-9.6) amplifications (Fig. 2B).

The two most recurrently amplified oncogenes in EAC, *KRAS,* and *ERBB2,* show stark differences in the proportions of alterations due to ecDNA and BFBs. In 82 tumors with *ERBB2* amplicons, 52% of tumors harbored an ecDNA and 30% harbored a BFB event. In contrast, 83

**Fig. 2 | Landscape of amplicons in 710 EAC. A** Frequency of genomic regions amplified in 710 EAC tumors and associated driver genes in chromosomes with amplicons. The height of peaks shows the number of patients with an amplification in the genomic region. Events were classified by Amplicon Architect into BFB, ecDNA, complex non-cyclic amplicons, and linear amplification. **B** Distribution of amplicon copy numbers affecting oncogenes in EAC. Individual points show each amplicon per patient. The recurrence of each oncogenic event is shown above each violin plot. **C** Regions amplified in *KRAS* amplicons in ecDNA and BFB events. Each horizontal line shows the genomic region amplified per patient. H3K27Ac[18] and gene annotations are shown below. The density plot (gray) shows the regions amplified aggregated across the cohort. Previously identified driver genes are highlighted in red. **D** Regions amplified in *ERBB2* associated amplicons in ecDNA and BFB events. Source data are provided as a Source Data file.

---

tumors with *KRAS* amplicons have a predominance of BFB events (63%) compared to ecDNA (30%). Compared to amplified regions spanning *KRAS* (Fig. 2C), the *ERBB2*-associated regions span multiple clusters of putative enhancer elements characterized by H3K27Ac (Fig. 2D), based on previously published ChIP-seq data[18]. Furthermore, the ecDNAs comprise a more focal genomic region of amplification (median size 410.8 Kb, IQR = 221.0–731.0 Kb) compared with a larger region with various amplicon sizes in the BFBs (median 719.5Kb, 30.9–1370.8 Kb). The regions amplified in cases with *ERBB2* ecDNA show an additional peak proximal to *JUP* (Fig. 2D), due to the co-amplification of the region when the *JUP* enhancers interact with the *ERBB2* locus (Fig. 2D) shown in a previous study with Hi-C data[19]. We identified seven tumors with ecDNA involving the *JUP* and nearby enhancers without *ERBB2* amplification (median CN = 9.06, IQR = 7.31-14.88), suggesting the *JUP* amplicons provide a selective advantage independent of *ERBB2* amplification (Supplementary Fig 1).

Next, we set out to identify associations between the regions amplified in BFBs and ecDNA and the presence of genetic elements and transcription factor binding sites. We modeled the frequency of regions amplified in 100 kb bins in the genome[20] and included annotations with replication timing, DNase I accessibility, and ChIP-seq from Encode[21] (H3K36me3, H3K27ac, K3K4Me3) and experimental data[18,22,23] (*GATA6* and *HNF4A*, H3K27Ac in tumors and cell lines). We identified an association between amplifications with late-stage replication timing in both BFB and ecDNA amplicons and an association of *HNF4A* binding sites and the presence of putative enhancers with H3K27Ac with ecDNA amplicons (Supplementary Table 3).

## De-novo assembly of long reads classifies complex amplicons into BFBs and ecDNA

Many of the amplicons in EAC are highly amplified, contain multiple segments, and span multiple chromosomes that are difficult to reconstruct using short-read sequencing-based methods. To overcome this difficulty, we re-sequenced nine tumor samples characterized by the presence of ecDNA and BFBs (where DNA is available) and three paired normal genomes using Oxford Nanopore long-read sequencing (30x coverage, N50 = 10-20 kbp). We carried out *de-novo* assembly[24] to reconstruct these complex amplicons and to classify each assembly graph into ecDNA and BFBs. We classified ecDNAs as assembly graphs that form a cyclic conformation (i.e., sequences that form a circular path back to the origin, Supplementary Fig. 2A–F) and BFBs that form a linear sequence with inversions and linear amplifications (Supplementary Fig. 2G–J). A total of 19 amplicons, including five BFB, nine ecDNA, and five complex non-cyclic events, were assembled, with a high concordance in the ecDNA classifications from AA, resulting in eight cyclic assembly graphs (Supplementary Table 4). We identified one tumor with the tandem duplicator phenotype (TDP) that resulted in false positives in AA predictions arising from low CN gains (Supplementary Fig. 2I–J).

Of the six patients with amplicons and long-read sequencing, we chose three patients with ecDNA arising from distinct mechanisms and affecting known oncogenes in EAC. We built molecular profiles based on genomic and clinical information (Fig. 3), and the other assembly profiles that illustrate similar mechanisms are shown in Supplementary Fig. 2B, F.

## Long read assemblies resolve complex amplicons and identify initiating processes

Patient 43 showed a TDP genomic profile[25] (based on SV signatures[26]), with a high number of low CN duplications (Fig. 3A). Of note, *ERBB2* ecDNA was present in both the BE sampled adjacent to the tumor (43B, CN = 7) and EAC (43T, CN = 41, estimated by Hatchet2.0[27]) collected from the same patient at the resection time point (Fig. 3B). The assemblies generated cyclic graphs (Fig. 3C) and a pairwise sequence alignment showed that both graphs share identical sequences, with the tumor having an additional segment containing keratin genes (chr17:38879471-39031761, Fig. 3A). The initiating event of the amplicon was due to a break in *CDK12*; followed by a duplication containing *ERBB2* (chr17:37663478-38206775) shared by both BE and tumor, that generated the circular ecDNA through an episomal mechanism[3]. The amplicon is likely to have originated in the BE stage and progressed to the tumor. When comparing the methylation profiles of the tumor, BE and normal squamous biopsies, we identified a DMR in the segment containing keratin genes (Fig. 3E). The DMR overlapped with H3K27Ac signal (proximal to *SMARCE1*) and multiple H3K27Me3 regions spanning *KRT222-KRT10* genes, corresponding to a putative enhancer and a heterochromatic region respectively (Supplementary Fig. 3A).

Patient 18 had a high number of mobile element insertions (MEI) based on the SV profile[11] (Fig. 3F). We identified a complex amplicon event linking an ecDNA containing *CCNE1* (chr19), a BFB spanning *ERBB2* (Chr17) and a translocation between chromosome 17 and 18 between *DLGAP1* and upstream sequences of *NEUROD2* (Fig. 3G-H). The *CCNE1* ecDNA consisted of sequences from three additional genomic regions with genes including *VASP, MARK4, CYP2F1,* and *CIC* (Fig. 3H). We observed a region of hypomethylation spanning *CCNE1*, overlapping with H3K27Me3 marked heterochromatic regions in the tumor, that is suggestive of an accessible chromatin structure[5] within the ecDNA compared to a panel of normal squamous tissue (Fig. 3I, Supplementary Fig. 3B-C). Compared to the cyclic *CCNE1* ecDNA, the BFB regions consist of foldback inversions containing *ERBB2* and *NEUROD2*, and these showed focal DMRs in enhancer regions with a lower fraction of reads hypomethylated compared to the *CCNE1* ecDNA (Fig. 3I, J, Supplementary Fig. 3C).

We devised a strategy to separate reads originating from the BFB and ecDNA in the overlapped sequences (Supplementary Fig. 3D) based on the clustering of hypomethylated reads[28] in the ecDNA[5] (Methods, Supplementary Fig. 3E). We separated reads based on methylation profiles and assembled a cyclical graph containing *CCNE1* (Fig. 3J) that showed uniform coverage of reads throughout the graph (Supplementary Fig. 3F). The refined assembly graph deconvoluted the structure of the *CCNE1* ecDNA (Fig. 3J) and demonstrated the ability to use the methylation state of reads to resolve highly complex assembly graphs (Fig. 3H). We applied this approach to additional ecDNA graphs and identified two tumors with a refined cyclic graph (Supplementary Fig. 3G and H)

Patient 139 showed a profile with extensive MEI in the tumor and harbored a *CDK6* ecDNA and *KRAS* BFB (Fig. 3K, L). The *CDK6* amplicon contained a germline LINE-1 (Long interspersed nuclear element-1) and somatic insertions linking segments of sequences from multiple chromosomes, spanning *CCND3, VEGFA* and *CDK6* in the amplicon (Fig. 3L–M, Supplementary Fig. 4A). The assembled amplicon graph contained two segments with LINE-1 sequences at a high CN (CN = 84)

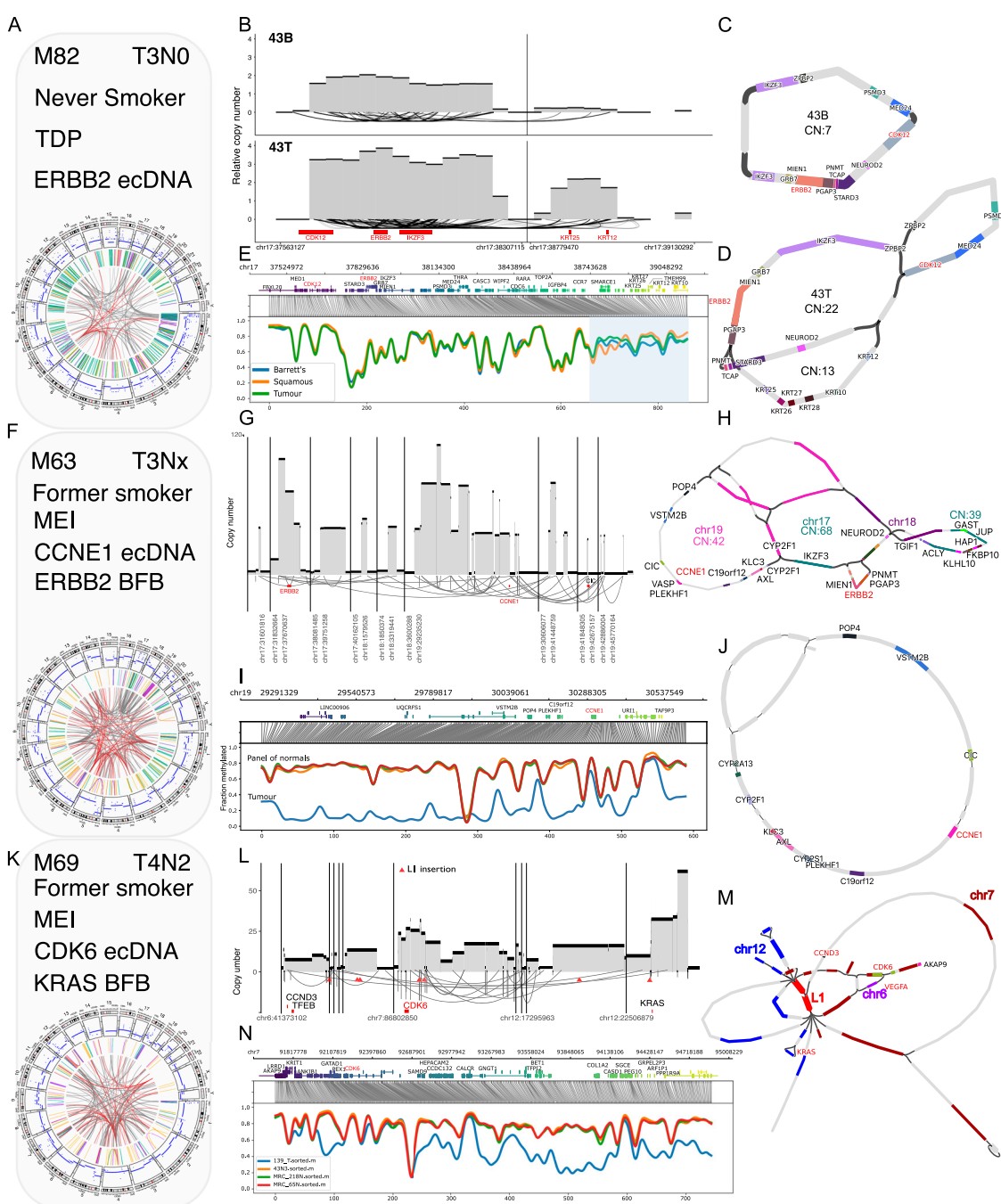

**Fig. 3 | Long-read assemblies resolve complex amplicons and identify amplicon-initiating processes. A–N** Three patient profiles with ecDNA were identified in their tumor genomes with clinical (gender M = male or F = female, age, T and N stage) and molecular features. **A** Profile of Patient 43 with the TDP phenotype and *ERBB2* ecDNA. **B** CN profiles of BE (43B) and tumor (43T) containing amplicons with breaks in *CDK12* and spanning *ERBB2*. **C** Assembly graphs of patient P43, 43B, and **D** 43T showing CN and position of genes amplified. 43T contains a segment of keratin genes (CN = 13) in addition to segments shared with 43B. **E** Methylation profiles of the amplified regions in 43T showing the fraction of reads methylated with gene annotations above. **F** Profile of tumor of patient P18 with a

*CCNE1* ecDNA and *ERBB2* BFB. **G** CN profile of a complex ecDNA and BFB event. **H** Assembly graph of complex amplicon spanning three chromosomes: chr17,18 & 19. **I** Methylation profiles of regions spanning *CCNE1* and **J** Assembly of *CCNE1* ecDNA deconvoluted based on hypomethylated reads. **K** Profile of tumor from Patient P139 driven by LINE-1 insertions. **L** CN profile of a complex amplicon containing a *CCND3* amplicon, *CDK6* ecDNA, and *KRAS* BFB. Arrows indicate LINE-1 insertions identified using TraFic[47] and TLDR[42]. **M** Assembly graph of amplicon included 2 segments containing LINE-1 sequences (CN = 84). **N** Methylation profile of LINE-1 containing amplicon.

and evidence of a somatic LINE-1 transduction downstream of the *CDK6* gene that was inserted in *CASC1*, upstream of *KRAS* (Fig. 3M CN = 173, Supplementary Fig. 4B). Repeatmasker annotation of the MEI sequences identified two LINE-1 sequences from L1HS and L1PA2 families previously unresolved using short read sequencing. We profiled the methylation status of the source and insertions of LINE-1 and

found hypomethylation in sites of the LINE-1 insertion (Supplementary Fig. 4A and B). To identify tumors in the 710 cohort with LINE-1 insertions near complex amplicons, we integrated TraFic MEI and AA calls and identified an additional 43 (6%) of tumors (Examples shown in supplementary Figs. 4C–E). Despite limitations in short-read sequencing to resolve LINE-1 insertions, we show that LINE-1 insertions are a

plausible mechanism that is associated with the origin of complex amplicons.

## Organoid models as preclinical models that recapitulate patient tumors

Next, we interrogated patient-derived organoids (Supplementary Table 5) with oncogenic amplicons to determine if they were suitable preclinical models for characterizing complex events, including ecDNA. Seventeen of twenty-four organoid cultures were found to harbor amplicons affecting nine recurrently altered oncogenes in EAC

(Fig. 4A). We did not find any amplicons in 7 organoids and their corresponding tumors and omitted them for further comparisons. We curated individual amplicon events between paired tumor and organoid, which showed that events were 94% concordant between the organoids and tissue, with 3 out of 45 (6%) events discordant (1 observed in the tumor, and not present in organoid, 2 events detected in the organoid but not in the primary tumor, Fig. 4A, Supplementary Table 6). Importantly, we observed that organoids captured the characteristics of amplicons in primary tissues, with stable CN profiles in BFBs throughout passages (Fig. 4B). We also observed higher CN

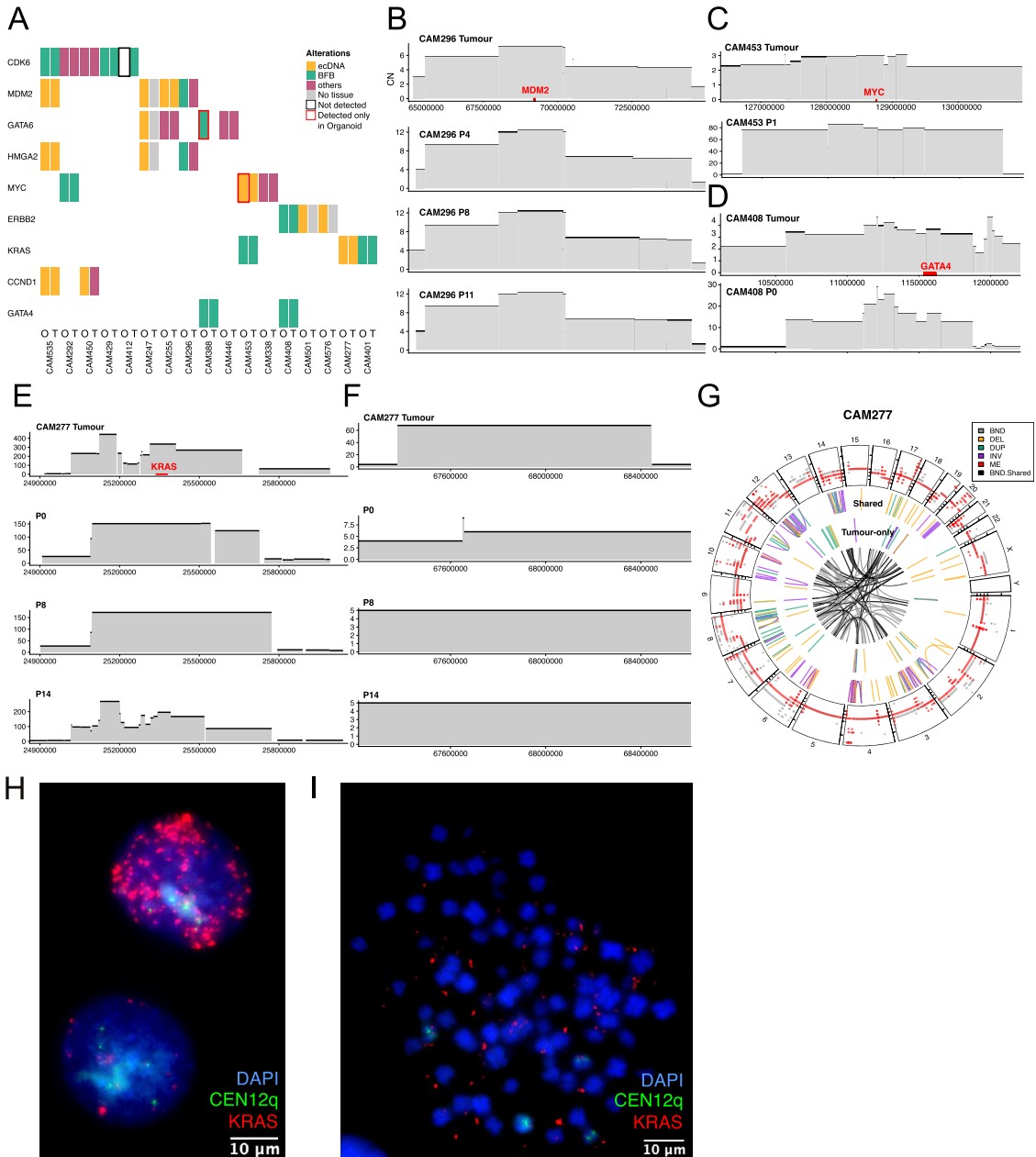

**Fig. 4 | Organoid models as preclinical models for characterizing complex amplicons. A** Oncoplot showing amplicons identified in organoid (O) and paired tumor (T) tissue from the same patient, classified by Amplicon Architect. **B** Copy number profile of a BFB event in the primary tumor and organoid CAM296 sequenced at passages (P) 4, 8 and 11. **C** Copy number profile of an ecDNA event detected only in the organoid CAM453 containing the MYC oncogene. **D** Copy number profile of a BFB event in CAM408 enriched in the organoid compared to the tumor. **E** Copy number profile of an ecDNA spanning *KRAS* in the tumor, CAM277 organoids at passage 0, 8, and 14. **F** Copy number profile of a

second ecDNA on chromosome 12 in CAM277 that diminished across passages. **G** Circos plot showing overlap of SV and CNA profiles of CAM277 (red) and primary tumor (gray). **H** Interphase FISH of CAM277 showing *KRAS* ecDNA against centromere labeling for chromosome 12 (CEN12q). **I** Metaphase FISH of organoid CAM277 with additional DAPI staining for DNA. 60X magnification was used for the interphase and metaphase FISH. The Metaphase FISH was carried with 2 replicates and shown with 10μm scale bars. Source data are provided as a Source Data file.

values in organoids in the absence of contaminating cells and a higher purity of tumor cells (Fig. 4B–D).

In addition, organoid cultures enabled a better classification of amplicon types for complex events such as ecDNA (CAM296, *MDM2* and *HGMA2* amplicon, Fig. 4A) and detection of ecDNA that are poorly represented in the primary tissue but detected in the organoid (Fig. 4C). In CAM453, a duplication event (chr8:126694685-130657526) was identified in both the tumor and organoid (CN 0.4 and 108.5 respectively) using GRIDSS-LINX[29], suggesting that the amplicon was present at a low CN in the tumor and clones harboring the *MYC* ecDNA expanded in the derived organoid (Fig. 4C).

We identified an organoid (CAM277) with an amplicon CN profile that differed from the primary tumor (Fig. 4E, F) and showed a clonal shift across passages in a previous study[14]. CAM277 showed a high overlap of SV events between organoid and tissue, with 54.3% of SV events overlapping and 19.6% of tumor SV events absent in the organoid (Fig. 4G). The large overlap of complex events suggested that clones harboring large-scale alterations present in the tumor were present in the organoid. We identified two separate ecDNA amplicons on chromosome 12, the first containing the *KRAS* oncogene and a second containing genes *CAND1*, *DYRK2*, and *IFNG-AS1* (Supplementary Fig. 5A, B). We carried out interphase and metaphase FISH on CAM277 to detect ecDNAs containing *KRAS* and identified cells with varying amounts of ecDNA in keeping with the stochastic inheritance (Fig. 4H, I). Hence, we deduce that changes in CN profiles in the *KRAS* locus (Fig. 4E) were due to the depletion of clones with ecDNA events after organoid derivation followed by the expansion of ecDNA-containing clones at passage 14. The converse occurred for the *IFNG-AS1* ecDNA that was at a high CN (CN = 60) in the tumor and diminished in the organoid at passages 0, 8, and 14 (Fig. 4F).

As several organoids harbored complex amplicons, we carried out long-read sequencing on three organoids (CAM277, CAM535, and CAM408) to assess our *de-novo* assembly-based method. We assembled these amplicons, compared to AA and AAClassify to discern between ecDNA and BFB events in the absence of contaminating cells and low cellularity, with three out of six ecDNA events predicted forming cyclic graphs and four out of four BFB events forming linear segments with inversions (Supplementary Table 6, Supplementary Fig. 6A–F). We curated the discrepant cases and found that all three cases had a CN of 10 or less, and the number of reads spanning the ends of the linear sequences was less than two, hence these cases were missed due to lower sequencing depth (Supplementary Figs. 6E, F).

**Single-cell sequencing in tandem with long read assembly allows for tracking of clonal shift in organoids**

One of the key aspects of ecDNA events is the ability to be passed on to daughter cells in a stochastic manner with the potential for clonal selection to go unchecked. However, the evidence supporting this process is currently sparse. The organoid model and single-cell DNA sequencing methods make tracing of this process more tractable.

To characterize the clonal shifts from the primary biopsy to the organoid across passages 4 and 15 in CAM277, we carried out bulk short-read sequencing of the normal squamous, tumor, and organoids and DLP+ single-cell sequencing at two time points[30]. UMAP clustering of 0.5Mbp segmented genomic scDNA copy profiles identified 4 subclonal populations (Fig. 5A) with clones A, B, and C being enriched at passage 15 and subclone D was enriched in passage 4 (Fig. 5B, C).

To quantify the ecDNA constructs on a single cell level, we used the assembly graphs of the ecDNA as a reference sequence (Fig. 5D) to map DLP+ reads per single cell and previously published bulk short-read WGS data[14] (Fig. 5E top panel). We used HMMCopy[31] to normalize the read counts, using GC content and mappability for each sequence, to obtain CN values for each segment. Most notably, BFB-associated sequences decreased between the tumor biopsy and across organoid passages, while ecDNA-associated sequences (e25) containing KRAS

and correlated sequences increased (Fig. 5E). We used the correlation between segments to identify at least two different *KRAS* ecDNAs present in the organoid. The first ecDNA was present in the tumor and P14 and a second ecDNA at P0 and P8 that had additional segments e1 containing *C12orf77* (chr12:251389380-25148653) and e12 containing *LMNTD1* mapping to chr12:25628038-25636598 (Figs. 4E, 5E). We used the normalized CN to estimate the copy number of each sequence per single cell clone and identified the presence of multiple possible ecDNA containing *KRAS* and *C12orf77* with high median absolute deviation (MAD) of the copies of ecDNA between cells. This recapitulated the stochastic distribution of ecDNA molecules present in individual cells as observed using FISH (Figs. 4H, 5F). The distribution of CN values for BFBs (e17) is shown to decrease in CN between passages (Fig. 5F). In addition, we identified a complex amplicon on chromosome 4 that decreased across passages (Supplementary Fig. 7A–B) and recapitulated the events on chromosome 12 based on the single cell CNV and SV events[32] (Supplementary Fig. 7C–D). Overall, the increase in median CN of the *KRAS* amplicon between passages points to the positive selection of clones containing the ecDNA, and the variation in CN values and segments amplified (Fig. 5F, G) demonstrates the stochastic nature of these ecDNA molecules.

## Discussion

In summary, we have shown that the highly prevalent amplification events in this tumor type are predominantly non-linear and complex, including ecDNA events. This high prevalence is due to the strong selective advantage conferred by those events harboring known oncogenes. However, we did not identify an association (p = 0.051) of the presence of amplicons (BFB or ecDNA) with poorer survival (Supplementary Fig. 8A). We identified that rearrangement processes in EAC such as tandem duplication and MEI are implicated in these amplicons. Of note, the presence of breaks in regions with nearby enhancer activity and transcription binding sites suggests a strong role of epigenetic regulation that results in novel chromatin interactions. We deduce that epigenetic regulation can both initiate the formation of these amplicons and result in *cis* or *trans* interactions with other regulatory elements. Interestingly, recent studies[19,33] have identified enhancer RNAs (eRNAs) at the *ERBB2-JUP* genomic loci identified in this study that have yet unappreciated roles in EAC pathogenesis. A recent study demonstrated the mechanism of estrogen receptor alpha binding in breast cancer leading to SVs and amplicons[20], which is a generalizable mechanism for the formation of complex amplicons, such as *ERBB2-JUP* amplicons, in EAC. After the formation of these amplicons, the evolutionary trajectory for clones shifts, often in favor of clones harboring these ecDNAs. This is observed in the organoid cultures that have a higher representation of ecDNA containing clones post organoid derivation and increased CN observed after passages in the single cell DNA sequencing (Fig. 5F, G).

The identification of initiating rearrangements and copy number changes in known amplicon regions may provide a useful biomarker for the early detection of EAC in the clinical setting. The differences and biases in genomic regions in the initiation of BFBs and ecDNA can be due to the sequence of the region or the presence of regulatory elements[20]. We detected hotspots with complex non-cyclic events affecting *GATA6*, *GATA4*, and *CCND1* that can have an underappreciated effect on EAC pathogenesis. Following the initiation event, CN gain leads to downstream effects including over-expression or gene regulation of nearby genes. Over-expression of oncogenes, based on the number of copies of a gene, is limited to the number of BFB cycles that occurred whereas the formation of ecDNA can bypass this limitation due to the stochastic inheritance of oncogenes in subclones. We have shown that the presence of ecDNA can be observed early in dysplastic BE[13] and in our study with P43 with the same *ERBB2* ecDNA in both the Barrett's and tumor biopsy. It may be possible to risk stratify BE patients according to evidence of any initiating events

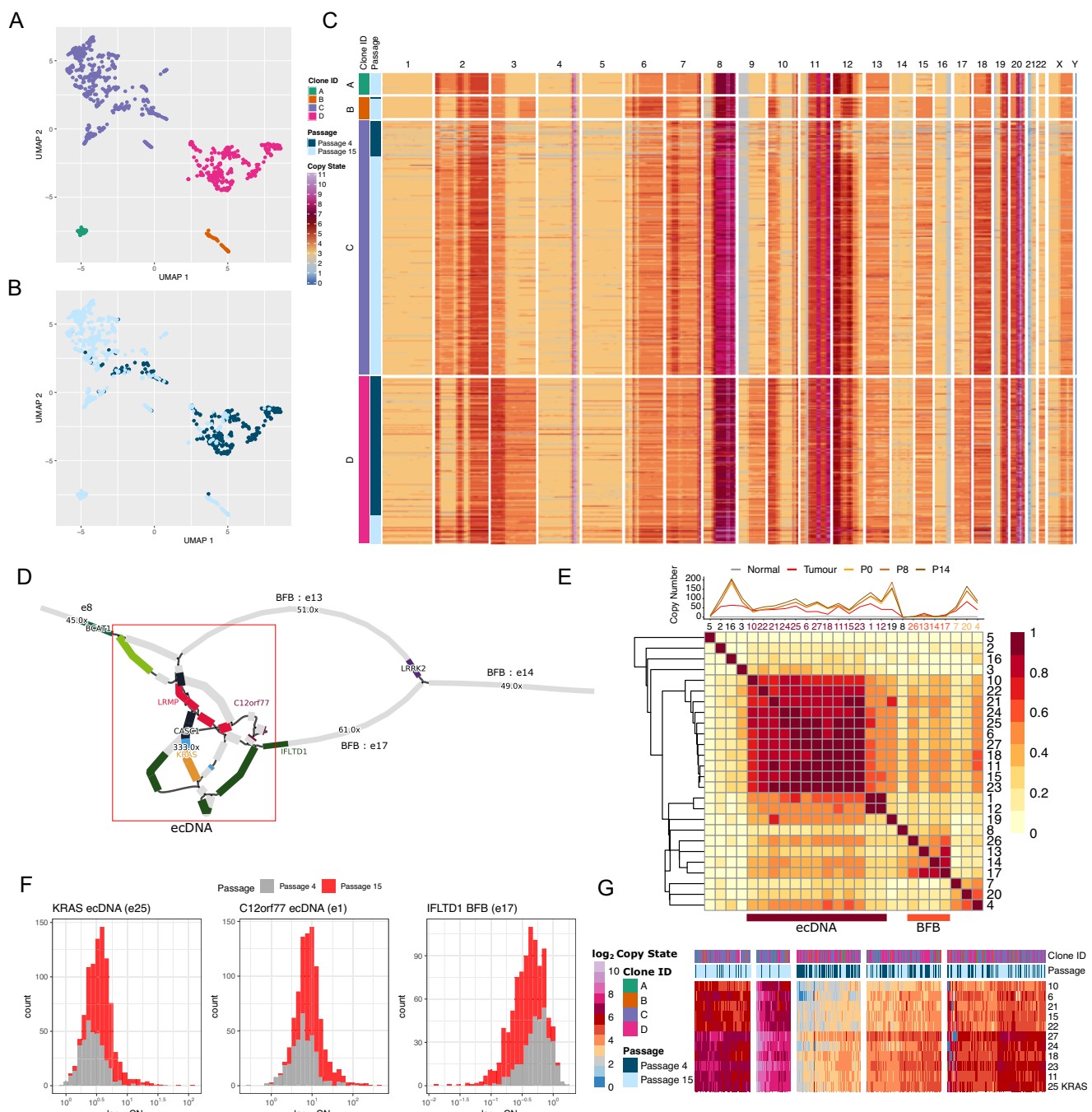

**Fig. 5 | Single-cell analysis disentangles complex amplicon and identifies ecDNA-containing clones.** **A**–**B** UMAP clustering of DLP+ sequenced single cells into subclonal populations in Passage 4 and 15. **C** Heatmap of the segmented genomic copy landscape in single cells across 2 passages in CAM277. Genomic regions are binned into 0.5Mbp bins. **D** Assembly graph of complex amplicon containing BFB sequences (edges 13, 14, and 17) and complex amplicon sequences (red box). **E** Line graph showing CN of bulk short-read WGS from normal squamous epithelium, tumor, and organoids at passage 0, 8, and 14 aligned to assembly graph

and a correlation matrix of scDNA-sequenced clone CN. Two clusters mapping to ecDNA sequences and BFB sequences were identified. **F** Distribution of ecDNA containing *KRAS* and *C12orf77* in individual single-cell clones at passages 4 and 15. Distribution of CN values in the *IFTLD1* BFB with no clonal shift shown as a comparison. A total of 354 cells for passage 4 and 580 cells for passage 15 were included in the analysis. **G** Heatmap showing CN values of scDNA clones mapped to assembly graph sequences. Edge 25 (e25) is the graph edge containing the *KRAS* sequence. Source data are provided as a Source Data file.

or amplified regions. We expect these amplicon events to be at a lower CN compared to the tumors and obscured due to the presence of multiple subclones in an earlier stage of the disease. Therefore, developing approaches to detect these events early in the pathogenesis of cancer is an area for further research.

We developed ecAssemble to carry out the *de-novo* assembly of these complex amplicons, by resolving complex structures with long

reads and employing methylation profiling to identify functional changes within these complex structures (Methods). In summary, we find that it is possible to use three types of information: 1) regions with high amplification, 2) the presence of assembled cyclic graphs and 3) differentially methylated regions (DMRs) with a large fraction of hypomethylation on enhancer and heterochromatic regions, to discern ecDNA events from BFB and other complex non-cyclic events.

The assembly-based approach can be used to deconvolute complex amplicons (with the integration of methylation information), generate reference sequences to better resolve and quantify ecDNA structures that have repetitive sequences, and use either bulk or single-cell genomic data to decipher the clonal dynamics in cells with complex rearrangements or amplicons. Especially in cases with highly complex structures like in the *CCNE1* amplicon in P18 (Fig. 3H,J), the use of methylation states resolved the ecDNA structure despite overlapping sequences with the BFB. We identified that the methylation patterns within ecDNA in our study varied between tumors. In addition, we compared the fraction of methylated reads between samples and regions with and without ecDNA amplicons in each tumor. Despite a small set of tumors, we observed that the methylation patterns are determined by the overall methylation levels in a tumor genome and the presence of large regions containing enhancers or heterochromatic marks (Supplementary Fig. 8B and C).

A limitation of the study was due to reads not spanning the multiple segments or the entire structure of ecDNA, so our ability to identify multiple ecDNA circles was limited. However, we envision that the use of ultra-long reads (in the megabase ranges) will allow for the resolution of multiple structures and provide a better estimation of the diversity of ecDNA in tumors. The assembly-based approach is also limited by the number of reads spanning ecDNA junctions to generate a complete graph. In this study, we opted to carry out whole genome long-read sequencing to generate a representation of all complex amplicons in each tumor instead of sequencing to focus on the validation of ecDNA events. Experimental[34] or sequencing strategies[35] to enrich ecDNA reads can address this limitation.

Our integration of long-read sequencing data with single-cell DNA sequencing allows for robust quantification of ecDNA in individual cells. This approach provides an alternative to microscopy-based methods[36] for ecDNA quantification in addition to a recently reported scRNA-based approach[37]. Most importantly, the clonal dynamics of individual clones and cells can be tracked across passages, in primary tissue-derived organoids, to identify the changes in the preponderance of different ecDNA and amplicons. It is possible to trace the lineages of cells according to their inherited genetic alterations and to model possible changes in clone fitness when more than one amplicon is present. The limitation of this model is that the in-vitro organoid model will post a different selective environment than in-vivo, especially without the constraints in the tumor microenvironment and other immune cell types. We foresee the integration of long-read and single-cell sequencing data to provide new sequencing-based tools to dissect changes in clonal dynamics due to complex amplicons with higher resolution and granularity, especially in experimental systems such as patient-derived organoids.

## Methods
### Study design, cohort and sequencing
This study complies with all relevant ethical regulations. The study was approved by the Cambridge South Research Ethics Committee (REC 07/H0305/52 and 10/H0305/1) and included written individual informed consent. EAC samples were obtained from surgical resections performed at Addenbrooke's Hospital and clinical information was collected following written informed consent as part of the OCCAMS study. Since all cases were selected based on having surgery and there were no samples taken from distant metastases in this cohort. Gender based analyses have not been done as EAC has a high male dominance and an analysis on female cancers would likely be underpowered given the available data.

A cohort of 710 esophageal adenocarcinoma patients with endoscopic and resection specimens were selected for study as part of the OCCAMS study. Patients were predominantly male (84.9%), with a median age of 66.8 years and stage T3 (Supplementary Table 1, Supplementary Data 1). Specimens were selected for Illumina sequencing

(100-150 bp, 50X coverage) if estimated purity > 70%, assessed through expert pathology review. Blood or normal squamous esophageal samples were used as a germline reference. Haematoxylin and Eosin (H&E) stained frozen tissue sections were reviewed by two independent pathologists for tumor cellularity and EAC tissue samples with ≥70% cellularity were selected for extraction, and sequencing reads were mapped using BWA-mem (V0.7.17).

### Structural variant calling on short read sequences
Structural variants were called using Manta[38] as previously reported[11], for the 710 short-read sequenced tumors. In addition, we carried out integrated CNV and SV calling using the GRIDSS-Purple-LINX[29] suite using default parameters, to allow for the comparison of SVs in the tumors and organoids. LINX annotation of complex clusters was used to further annotate complex non-cyclic events to identify the pattern SV types in each cluster.

### Identification and classification of amplicon events
Copy number segments were called using CNVKit[39] v0.9.8 and regions of amplifications of size 50 kb, copy number > 4.5 were used as input for the identification of amplified regions and reconstructed using Amplicon Architect v.1.2[17]. The classification of amplicons into ecDNA, BFB, linear amplifications, and complex non-cyclic events was performed using Amplicon Classifier v0.4.13[13].

### Oxford Nanopore sequencing and data processing
DNA from fresh frozen tissue was extracted using the QIAGEN Genomic-tip 500/G kit, sheared using a g-TUBE, and adapters were ligated using the LSK109 Ligation sequencing kit. Sequencing was carried out using a PromethION with R9.4 flowcells and base-called using Guppy 5.0.11 in the high accuracy (HAC) mode. Reads were aligned using Minimap2 (v2.26-r1175). Methylation calling was carried out using Megalodon v2.4.2. CNV calling was done using QDNAseq v1.18.0[40] and SV calling was carried out using Sniffles2 v2.06[41].

### Amplicon assembly and functional annotation
We defined the amplified regions for assembly using the Amplicon architect predictions based on short-read data from the same biopsies. To carry out *de-novo* assembly, we extracted reads mapping to the amplified regions and used Flye v2.9.3-b179[24] to assemble the amplicons. To identify amplified genomic features and mechanisms generating breakpoints in these complex amplicons, we annotated each graph with the genomic coordinates of the reference genomic sequence mapping to each sequence; gene annotations present on each segment, evidence of rearrangements at the locus and methylation profiles of each segment. We used these features to identify the set of oncogenic genes within the amplicons; functional elements such as enhancers and regulatory regions, and rearrangement breakpoints in these circles to provide possible insight into the mechanisms leading to the formation of the circular amplicons and their biological consequences. Previously published ChIP-seq data from tumors and cell lines[18] and esophagus cell line E079 from Epigenome Roadmap[21] were used for additional annotations of regulatory elements including enhancer and heterochromatin elements. TLDR v0.1[42] was used to identify LINE-1 elements in the long-read sequences.

To resolve the complex amplicon in P18, we developed ecAssemble (https://github.com/fitzgerald-lab/ecAssemble, Supplementary Data 2) to carry out an assembly of the entire amplicon based on clusters of methylated reads. We generated 10Kb windows spanning the amplicon sequences and clustered the long reads using the CVLR v0.1[28] tool with the number of clusters = 2. Using the reads from each hypomethylated cluster, we re-assembled the filtered reads using Flye to generate the refined assembly map in Fig. 3J and Supplementary Fig. 3G–H.

## Sample collection, organoid derivation, and culture

Half of the EAC patient tissue samples were prepared for organoid derivation while the other half were snap-frozen using liquid nitrogen and stored at −80 °C until used for genomic profiling. Organoid derivation and culture were performed by first washing the tumor samples using Phosphate Buffer Solution (PBS) before being minced using a scalpel and incubated using collagenase II for 1–2 hours at 37 °C[14]. The incubated mixture was filtered using a 70-µM filter to remove undigested fragments. The filtered cell suspension was then centrifuged at 300–400 g for 2 mins and resuspended and centrifuged again twice to remove debris and remaining collagenase. The snap-frozen tissue was stained with Haematoxylin and Eosin and the cellularity of the sample was reviewed by two pathologists independently. Tissues with ≥70% cellularity underwent DNA and RNA extraction using the AllPrep Kit (Qiagen) and were sequenced on paired-end Illumina sequencing to a depth of 30x. Blood or normal squamous esophageal samples were selected as germline reference samples. The organoids that showed robust growth after passaging and had whole genome sequencing were all included in this study.

To passage the organoids, the basement membrane matrix (Cultrex BME RGF type 2 (BME-2), (R&D Systems)) was dissociated, and the organoids were collected. Following the addition of TrypLE (Invitrogen) the suspension was incubated at 37 °C for approximately 20 min. A vigorous manual shake would ensue at regular intervals (5 min) and upon completion, the suspension was centrifuged at $300–400 \times g$ for 5 min. The resultant cell pellet was re-suspended in BME-2 and plated as 10–15 µl droplets in a 6-well plate. After allowing the BME-2 to polymerize, IntestiCult™ Organoid Growth Medium (StemCell Technologies) supplemented with Primocin (1 mg/mL, InvivoGen) and 10 µM Y-27632 (TOCRIS) were added and cells were incubated at 37 °C. Organoid growth medium was refreshed every 2-3 days.

## Fluorescent in situ hybridization

Fluorescent in situ hybridization (FISH) assays were performed using non-diagnostic KRAS/CEN12q (Abnova) probes. To capture ecDNA, metaphase FISH was performed on EAC organoid cultures and followed standard cytogenic procedures for harvesting, fixation (3:1 methanol: acetic acid solution), and slide formation.

All FISH pre-treatment and hybridization steps were performed by the Department of Histopathology, Cambridge University NHS Foundation Trust, and Cancer Research UK Cambridge Center and followed the manufacturer's instructions. All slides were reviewed by a senior molecular pathologist and scored using current EAC diagnostic guidelines where applicable (e.g., HER2).

## Single Cell Genomic Sequencing

Single-cell DNA sequencing has been performed on previously characterized organoids[14]. Organoids from passage 4 and passage 15 were frozen in Recovery Cell Culture Freezing Medium (Thermo #12648010) and processed at BCCRC as described[30]. Briefly, samples were gently thawed, and single cells were isolated from frozen organoid suspension using Trypsin treatment. Single cells were spotted using a cellenONE (Scienion) instrument. Subsequently, libraries were constructed using previously described protocol[30] and approximately 1000 cells were sequenced on an Illumina HiSeq 2500 instrument.

## DLP+ Data Processing

Analysis of DLP+ data relies on a slightly modified version of the published DLP+ pipeline[43]. In brief, starting with paired FASTQ data, we trimmed reads using TrimGalore v0.6.6[43] and checked FASTQ quality using FASTQC v0.11.9[44]. The trimmed reads are then aligned to the human genome (GRCh37) using BWA-mem[45]. Along with the reference sequence, each cell is also screened for contamination using FastQScreen v0.14.0[46], producing a finalized BAM alignment file for each cell. These per-cell BAM files were merged to produce one BAM file for each experimental condition specified for the run. Aligned files are then run through the somatic copy number pipeline using the tool HMMcopy v0.0.23[31] in 500kbp bins and with GC-bias correction. An overall quality score is then computed for each cell based on an 18-feature random forest classifier trained on a large manually curated dataset[30]. The resulting cells were filtered for quality, leaving high-quality (> 0.7 quality score) cells from both passage 4 and passage 15 to merge. Clustering on the merged data was conducted via HBDScan to identify clonal populations.

## Quantification of ecDNA segments and structures

To quantify ecDNA segments on both bulk and single-cell short read sequencing, we aligned reads mapping in the amplified regions to the assembly graphs using MiniGraph (v0.20) and generated paf alignment output. Next, we calculated read counts per 10 kb bins and normalized the read counts using HMMCopy v0.0.23[31], adjusting for mappability and GC content. Lastly, we generated the CN values by dividing the normalized read counts by diploid segments identified in the assembly graph and calculated the median CN of each segment.

To identify segments that are associated with the ecDNA BFB structures, we calculated a correlation matrix based on scDNA CN values for each cell and identified clusters that have correlated CN values. Several segments (e1, e16, e19, e20) show a high CN value due to being shared segments between different conformations of ecDNA and other complex structures. scDNA SVs were called using deStruct v0.4.22[32] and genotyped at passages 4 and 15. Clonal differences in CNV and SVs were identified based on bins with the highest variance in CN and variance in read counts for each SV event detected.

## Reporting summary

Further information on research design is available in the Nature Portfolio Reporting Summary linked to this article.

# Data availability

The sequencing data generated in this study have been submitted to the European Genome-phenome Archive (EGA; https://ega-archive.org/) under the accession numbers EGAD00001007785 and EGAD00001006083 respectively. The raw sequencing data are available under restricted access due to data privacy laws for sensitive controlled genomic data; access can be requested to the ICGC Data Access Compliance Office as described here: https://docs.icgc-argo.org/docs/data-access/daco/applying. Applicants must be affiliated with a legal entity and submit a project summary that conforms with policies concerning the purpose of the research, protection of the donors and security of the data. Once the application has been submitted, the ICGC DACO committee will review your application and you will hear back within ten business days. Access to the controlled data will be granted for a period of two years. Processed data to reproduce Fig. 2A is available from Zenodo (https://zenodo.org/records/10775258). Genome annotations with replication timing, DNase I accessibility, and ChIP-seq from Encode[21] (H3K36me3, H3K27ac, H3K4Me3, https://www.encodeproject.org/) and experimental data[18,22,23] (GATA6, and HNF4A: https://www.ncbi.nlm.nih.gov/pmc/articles/PMC6499311/bin/supp_gr.243345.118_Supplemental_Table_S3.xlsx, KLF: https://cdn.elifesciences.org/articles/57189/elife-57189-supp5-v2.xlsx, H3K27Ac: https://www.ncbi.nlm.nih.gov/pmc/articles/PMC8108390/bin/NIHMS1695582-supplement-Supplementary_Tables.xlsx in tumors and cell lines) were used for lasso regression. Previously published ChIP-seq data from tumors and cell lines[18] and esophagus cell line E079 from Epigenome Roadmap[21] (https://egg2.wustl.edu/roadmap/web_portal/index.html) were used for additional annotations of regulatory elements including enhancer

and heterochromatin elements. Previous short-read sequencing data of organoid[14] were used to identify clonal shifts (https://www.ebi.ac.uk/ega/datasets/EGAD00001004007). Source data are provided with this paper. The remaining data are available within the Article, Supplementary Information, or Source Data file. Source data are provided with this paper.

## Code availability

The code for ecAssemble is available from https://github.com/fitzgerald-lab/ecAssemble https://doi.org/10.5281/zenodo.10708121.

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

## Acknowledgements
The laboratory of R.C.F is supported by a Program Grant from the Medical Research Council (MR/W014122/1). This work was supported by Cancer Research UK (A15874, A22720, A22131). SAZ is supported by a Gates-Cambridge Trust scholarship. We thank the Human Research Tissue Bank, which is supported by the UK National Institute for Health Research (NIHR) Cambridge Biomedical Research Center, from Addenbrooke's Hospital. This research was supported by the NIHR Cambridge Biomedical Research Center (NIHR203312). The views expressed are those of the authors and not necessarily those of the NIHR or the Department of Health and Social Care. S.T. and K.N.O. were funded by the MacMillan Family Foundation (60210014) as part of the MacMillan Center for the Study of the Non-Coding Cancer Genome at the New York Genome Center including B.W.'s studentship.

## Author contributions
A.W.T.N. and R.C.F. designed the study and supervised the analyses with K.N.-O. and S.T. A.W.T.N., D.P.M., B.W., S.A.Z. and E.B. carried out the computational analyses. D.P.M., X.L. and A.M. carried out the experiments. O.G. and R.T.H. carried out the FISH staining of the organoid models. G.D. processed the sequencing data by aligning and generating the variant calls. A.M.R., N.G. and A.B. coordinated the data collection. A.M. A.B. and X.L. generated the long-read sequencing libraries. S.J. carried out an analysis of Barrett's esophagus cases. K.N.-O., S.A. generated the single-cell libraries and B.W. carried the computational analyses. A.W.T.N., R.C.F. and K.N.O. wrote the manuscript with contributions from all other authors. All authors read and approved the manuscript.

## Competing interests
R.C.F is named on patents related to Cytosponge and related assays which have been licensed by the Medical Research Council to Covidien GI Solutions (now Medtronic) and is a co-founder of CYTED Ltd. These are not directly involved in the topic of this paper. R.C.F. has received consulting and/or speaker fees from Medtronic, Roche, and Bristol Myers Squibb. The remaining authors declare no competing interests.

## Additional information

-Statistics and reproducibility

## Oesophageal Cancer Clinical and Molecular Stratification (OCCAMS) Consortium

Rebecca C. Fitzgerald [1] ✉, Paul A. W. Edwards[1,2], Nicola Grehan[1,5], Barbara Nutzinger[1], Aisling M. Redmond[1], Christine Loreno[1], Sujath Abbas[1], Adam Freeman[1], Elizabeth C. Smyth[5], Maria O'Donovan[1,5], Ahmad Miremadi[1,5], Shalini Malhotra[1,5], Monika Tripathi[1,5], Calvin Cheah[1], Hannah Coles[1], Curtis Millington[1], Matthew Eldridge[2], Maria Secrier[2], Ginny Devonshire [2], Sriganesh Jammula[2], Jim Davies[12], Charles Crichton[12], Nick Carroll[13], Richard H. Hardwick[13], Peter Safranek[13], Andrew Hindmarsh[13], Vijayendran Sujendran[13], Stephen J. Hayes[14,15], Yeng Ang[14,16,17], Andrew Sharrocks[18], Shaun R. Preston[19], Izhar Bagwan[19], Vicki Save[20], Richard J. E. Skipworth[20], Ted R. Hupp[21], J. Robert O'Neill[13,20,21], Olga Tucker[22,23], Andrew Beggs[22,24], Philippe Taniere[22], Sonia Puig[22], Gianmarco Contino[22], Timothy J. Underwood[25,26], Robert C. Walker[25,26], Ben L. Grace[25], Jesper Lagergren[17,27], James Gossage[17,28], Andrew Davies[17,28], Fuju Chang[17,28], Ula Mahadeva[17], Vicky Goh[28], Francesca D. Ciccarelli[28], Grant Sanders[29], Richard Berrisford[29], David Chan[29], Ed Cheong[30], Bhaskar Kumar[30], L. Sreedharan[30], Simon L. Parsons[31], Irshad Soomro[31],

Philip Kaye[32], John Saunders[14,31], Laurence Lovat[32], Rehan Haidry[32], Michael Scott[33], Sharmila Sothi[34], Suzy Lishman[2,35], George B. Hanna[36], Christopher J. Peters[36], Krishna Moorthy[36], Anna Grabowska[37], Richard Turkington[38], Damian McManus[38], Helen Coleman[38], Russell D. Petty[39] & Freddie Bartlet[40]

[12]Department of Computer Science, University of Oxford, Oxford OX1 3QD, UK. [13]Cambridge University Hospitals NHS Foundation Trust, Cambridge CB2 0QQ, UK. [14]Salford Royal NHS Foundation Trust, Salford M6 8HD, UK. [15]Faculty of Medical and Human Sciences, University of Manchester, Manchester M13 9PL, UK. [16]Wigan and Leigh NHS Foundation Trust, Wigan, Manchester WN1 2NN, UK. [17]Guy's and St Thomas's NHS Foundation Trust, London SE1 7EH, UK. [18]GI science centre, University of Manchester, Manchester M13 9PL, UK. [19]Royal Surrey County Hospital NHS Foundation Trust, Guildford GU2 7XX, UK. [20]Edinburgh Royal Infirmary, Edinburgh EH16 4SA, UK. [21]Edinburgh University, Edinburgh EH8 9YL, UK. [22]University Hospitals Birmingham NHS Foundation Trust, Birmingham B15 2GW, UK. [23]Heart of England NHS Foundation Trust, Birmingham B9 5SS, UK. [24]Institute of Cancer and Genomic sciences, University of Birmingham, Birmingham B15 2TT, UK. [25]University Hospital Southampton NHS Foundation Trust, Southampton SO16 6YD, UK. [26]Cancer Sciences Division, University of Southampton, Southampton SO17 1BJ, UK. [27]Karolinska Institute, Stockholm SE-171 77, Sweden. [28]King's College London, London WC2R 2LS, UK. [29]Plymouth Hospitals NHS Trust, Plymouth PL6 8DH, UK. [30]Norfolk and Norwich University Hospital NHS Foundation Trust, Norwich NR4 7UY, UK. [31]Nottingham University Hospitals NHS Trust, Nottingham NG7 2UH, UK. [32]University College London, London WC1E 6BT, UK. [33]Wythenshawe Hospital, Manchester M23 9LT, UK. [34]University Hospitals Coventry and Warwickshire NHS, Trust, Coventry CV2 2DX, UK. [35]Peterborough Hospitals NHS Trust, Peterborough City Hospital, Peterborough PE3 9GZ, UK. [36]Department of Surgery and Cancer, Imperial College, London W2 1NY, UK. [37]Queen's Medical Centre, University of Nottingham, Nottingham, UK. [38]Centre for Cancer Research and Cell Biology, Queen's University Belfast, Belfast BT7 1NN, Northern Ireland. [39]Tayside Cancer Centre, Ninewells Hospital and Medical School, Dundee DD1 9SY, UK. [40]Portsmouth Hospitals NHS Trust, Portsmouth PO6 3LY, UK.

