## [Peer Review File · Nature Communications]

Disentangling oncogenic amplicons in esophageal adenocarcinomaReviewers' Comments:

Reviewer #1:

Remarks to the Author:

The paper describes the analysis of oncogenic amplifications in a cohort of 710 Esophageal adenocarcinoma (EAC) patients. Using short-read sequencing, the authors identified frequent ecDNA and BFB amplifications of known oncogenes and reconstructed the amplicon structures. In addition to short-read sequencing, the authors analyzed a subset of samples with long reads, which allowed to resolve the architecture of complex amplicons, and revealed the different methylation profiles of ecDNA. Analysis of patient-derived organoids showed high consistency with the primary samples, although in some cases of clonal selection was observed.

The manuscript provides interesting insights into the mechanisms of ecDNA and BFB amplifications in EAC, which potentially extends to other cancer types. The authors convincingly demonstrate the utility of new technologies (long reads and single-cell sequencing, patient-derived organoids) for studying oncogenic amplifications. The manuscript is well-written and complete. A few detailed comments are below.

Major:

1. We were not able to find the information about patient treatment. For example, treatment type and duration may affect the clonal selection. Were there any correlations between treatment, risk factors, amplification status and survival, as it was reported by previous studies?
2. Long-read methylation calls were used to disentangle amplicon assembly graphs. Did the authors observe that most of the amplicons classified as ecDNA were unmethylated? Figure 2E suggests that this may not always be the case.
3. In the cases with decreased BFB CN prevalence in organoids, it would be interesting to see if this was a result of aneuploidy, since BFB cycles may lead to chromothripsis or LOH.
4. Supplementary table 1 provides a summary-level information of the cohort. Is it possible to add information about individual patients, e.g. amplification status, treatment, smoking status, race, etc? In addition, it would be more convenient if the tables were in excel format, rather than pdf.
5. The relevant custom code / pipelines should be made available via a public software repository

Minor:

1. In the summary figure, it would be informative to add the number of samples that were subjected to each sequencing method.

Reviewer #2:

Remarks to the Author:

"I co-reviewed this manuscript with one of the reviewers who provided the listed reports. This is part of the Nature Communications initiative to facilitate training in peer review and to provide appropriate recognition for Early Career Researchers who co-review manuscripts."

Reviewer #3:

Remarks to the Author:

The manuscript by Alvin et al. reported important insights into the Complex amplifications of EAC based on WGS and long-read sequencing, especially for ecDNA and BFB. The WGS of 710 patients comprehensively characterized the high copy number amplification mechanisms of the EAC genome. In addition, the author conducted long-read sequencing on 9 EAC patients and used de-novo assemblies and DNA methylation to reconstruct complex ecDNA and BFB amplicons, effectively solving the problem of constructing complex chromosome rearrangements in the short-read sequencing.

Finally, the author integrated the patient-derived organoids and single-cell DNA sequencing to explore the ecDNA clonal dynamics of EAC tumor evolution.

In summary, they present a series of bioinformatic works to decode the characteristics of these events using methylation, long sequencing reads, as well as single-cell sequencing data. They found that complex amplicons are already present in the early stage, and confirmed the genomic consistency between tumor and patient-derived organoids. They also illustrate the importance of using long-read and single-cell DNA sequencing to track EAC evolution. Although some of the conclusions are expected, they did provide insights into the complex rearrangement of different perspectives. In addition, some concerns need to be addressed:

Major comments:

1. The author analyzed the high copy number amplification mechanisms in EAC, including ecDNA, BFB, Complex non-cyclic and Linear amplification. In previous studies on structural variations, it has been mentioned that complex tandem repeats can lead to high copy number amplification (PMID: 32025012, 33007263, 36272974). Therefore, can the author refine the classification of this part and analyze their impact on oncogenes? Because complex tandem duplications play an important role in the high copy number amplification and regulating the expression of driver genes.
2. The author assembled high copy number amplicons of 9 EAC patients through long-read sequencing. The manuscript mentioned the high concordance in the results of ecDNA between AA and de-novo assembly. However, the author did not provide clear results of other complex amplification events. Is there any difference in these results? If there are differences, what are the reasons for them?
3. When analyzing the amplification of 9 long-read sequencing data, the author only used 3 paired normal genomes as controls. How were these three EAC patients selected? Why didn't you do it all? More details are needed to introduce the matched normal samples. The abundance of a mobile element or L1 sequence near ecDNA might be germline events.
4. For derived organoids, the manuscript mentions that the amplicon events are highly concordant between paired tumor and organoid (Line 264-266). Previous study has shown that some EAC patients have intratumoral heterogeneity (PMID: 31949146). How can the author avoid the troubles caused by intratumoral heterogeneity? The results show that there seems to be no impact of intratumoral heterogeneity, Are there specific requirements when selecting tumors? Among them, an organoid (CAM277) with an amplicon CN profile that differed from the primary tumor, and may heterogeneity also be one of the reasons?
5. The findings tend to be descriptive. Many observations are obtained through sporadic samples with long-read sequencing, but it seems difficult to get a solid conclusion. As 710 tumors are sequenced, is there any way to further validate the presence of mobile element insertion as a driver of complex amplicons' origin?

Minor comments:

1. Line 79, Figure 1 is not well-organized, as it seems that there are two Figure 1 in the manuscript.
2. Line 101, did the author notice that many amplicons contain the enhancer of oncogenes only, such as gene MYC and KLF5. Thus, these amplicons should be considered if omitted.
3. Line 167 Lack of legend for A Graph
4. Line 181. Again, are there any matched normal samples subjected to long-read sequencing for these three patients?
5. As the accuracy and sensitivity of SVs are the foundation of this study, a detailed method for SV calling for both short- and long-reads should be shown.
6. Line 230 Figure 3J annotated error
7. Please label chromosome numbers in Figure 3B-F
8. Please describe the legend of Figure 3H-I in more detail.
9. It is best to draw the copy number of ERBB2 in Supplementary Figure 1
10. Please consider labeling the H3K27Ac signal and the H3K27Me3 regions in the DMR region(Figure 2E)
11. Line 277 GRIDSS-LINX needs to annotate literature sources. In addition, please add the

description of GRIDSS-LINX to the method.

12. Line 529, the title of the section 'Availability of data and materials' should be bold.

Reviewer #4:

Remarks to the Author:

In this manuscript, the authors explore the genomic distribution and structure of gene amplification structures as extrachromosomal DNA and BFBs in esophageal adenocarcinoma. Overall, this study exhibits a robust foundation, as the results are substantiated by a diverse range of experimental and analytical data. Notably, the application of Nanopore sequencing, patient-derived organoids and single cell analysis for investigating extrachromosomal DNA and BFBs yields intriguing findings. The use of Nanopore sequencing to validate ecDNAs and BFBs and the discovery of potential associations with hypomethylation states are significant findings. Furthermore, their exploration of these amplicons in tumor organoids and the tracking of their dynamics in serial passages contribute to uncovering their origins and functions. Collectively, these findings position this manuscript as a relatively novel and comprehensive study in the field. While the data processing is generally robust, it would be beneficial to provide more detailed information in the methods section and supplementary tables.

However, in several instances, the manuscript appears somewhat unrefined and requires further polishing.

1. The analysis initially focuses on a '710 Cohort.' However, there appears to be no description provided for this dataset. It's important to clarify whether this omission is deliberate for the purpose of this study or if the same dataset has been utilized in previous projects. Exploring the associations between extrachromosomal DNA (ecDNA) and their clinical implications, as well as examining SNPs, holds valuable insights that could benefit the broader scientific community.
2. In Figure 1A, it appears that KRAS has a peak dominated by BFB, which does not align with the observations presented in Figure 1C.
3. As emphasized by the authors, it is noteworthy that BFB, ecDNA, and CNCs exhibit biases in genome regions. Therefore, it is advisable to include discussions and results regarding their distinctions in cellular functions and potential clinical implications.
4. In Figure 2B, it is observed that these two samples exhibit a relative copy number of 2-3, which does not appear to be indicative of extrachromosomal DNA (ecDNA). Furthermore, their methylation patterns closely resemble those of normal samples.
5. As in the BE (normal) sample, in the majority of cases, their copy numbers remain within the normal range. It is indeed surprising to detect a copy number (CN) and even an extrachromosomal DNA (ecDNA) with a copy number of 7.
6. Numerous graphs depicting ecDNA and BFBs, such as those found in Figure 2 and others, raise the question of whether these representations reflect the actual complexity of DNA structures that form and replicate within cells, or if they are rough assemblies of individual amplicon structures?
7. The second section of the results appears to be quite lengthy. It might be beneficial to consider dividing it into two separate sections for improved organization.
8. The association between the hypomethylation state and ecDNA reads appears to be underaddressed. Is this conclusion widely applicable and firmly established, or does it apply only to specific samples? The graph in Figure 2J presents a promising analysis; it might be worthwhile to conduct similar analyses for other samples as well.
9. In Figure 3A, the MYC seem to be detected also in tumor.
10. In Figure 3C, it appears that the organoids may have lost some of the amplicon details. Could this be attributed to changes in the amplicon structure within such a small genome region?
11. In Figure 3E, it's noteworthy that the tumor exhibits a higher copy number (CN) than the pure organoid, and this requires further explanation. Additionally, the sudden reappearance of the initial ecDNA structure in P14 appears suspicious. It is more likely that P0 and P8 did not fully detect the amplicon in detail and may have missed them.

12. In Figure 4F, the detection of the loss of a BFB amplicon is noteworthy. Given the availability of results from patient-derived organoids (PDOs) across multiple passages, it would be beneficial to conduct additional analyses to explore the evolutionary patterns of various types of amplicons.
13. Line 230, is it Supplement Figure 3J?

Reviewer #5:

Remarks to the Author:

In this manuscript, Ng, Fitzgerald, and colleagues present a study on oncogenic amplicons in oesophageal adenocarcinoma. They re-examine OCCAMS WGS data using AmpliconArchitect to describe the landscape of amplicons in OAC. They then use long read assemblies to study more complex amplicon structures in a subset of samples. Finally, they use organoids derived from OAC tissue to study the evolution of oncogenic amplicons during organoid culture and passage. While largely observational, this study uses cutting edge techniques to study the impact of amplicons on OAC tumorigenesis and act as a proof-of-concept for further study.

Major comments:

1. How were those 24 organoids chosen?
2. What happened to the data from the 7 organoids where you did not find amplicons with recurrently altered OAC oncogenes? For those 7 organoids, did the primary tissue have similar amplicons or did it have amplicons with recurrently altered OAC oncogenes? Did these 7 organoids recapitulate the primary tissue otherwise? Along those lines, does this imply that 30% (7/24) of organoids/tumours do not have amplicons as a mechanism of tumour propagation?
3. Depending on above answer, the 94% concordance reported does not take into account the 7 removed organoids.
4. For Figure 3A, what passage were the organoids taken to compare to the tumour?
5. Given stochastic inheritance and the multiple clones demonstrated by your scDNAseq, how do you think the heterogeneity of the organoids in culture at a particular passage affects the interpretation of the organoid data? Could the detection of ecDNA poorly represented in primary tissue be a result of de novo clones?
6. The authors demonstrate that organoids will clones harboring ecDNA expand during culture due to a competitive advantage. The authors suggest that this may allow for the modelling of clone fitness in vitro. I think that a potential downside is that the in vitro environment poses a different selective environment than in vivo and may not serve as an accurate representation of the in vivo evolution. This might be interesting in the discussion.

Minor comments:

1. How did you use the 3 paired normal genomes for Nanopore?
2. Some detail on how the organoids were harvested for WGS, depth of sequencing, etc, in the methods would be helpful.

REVIEWER COMMENTS

Reviewer #1, expertise in long-Read DNA Sequencing and extrachromosomal DNA (Remarks to the Author):

The paper describes the analysis of oncogenic amplifications in a cohort of 710 Esophageal adenocarcinoma (EAC) patients. Using short-read sequencing, the authors identified frequent ecDNA and BFB amplifications of known oncogenes and reconstructed the amplicon structures. In addition to short-read sequencing, the authors analyzed a subset of samples with long reads, which allowed to resolve the architecture of complex amplicons, and revealed the different methylation profiles of ecDNA. Analysis of patient-derived organoids showed high consistency with the primary samples, although in some cases of clonal selection was observed.

The manuscript provides interesting insights into the mechanisms of ecDNA and BFB amplifications in EAC, which potentially extends to other cancer types. The authors convincingly demonstrate the utility of new technologies (long reads and single-cell sequencing, patient-derived organoids) for studying oncogenic amplifications. The manuscript is well-written and complete. A few detailed comments are below.

Major:

1. We were not able to find the information about patient treatment. For example, treatment type and duration may affect the clonal selection. Were there any correlations between treatment, risk factors, amplification status and survival, as it was reported by previous studies?

We thank the reviewers for their comment, the OCCAMS cohort predominantly consist of primary tumours sequenced before treatment in order to avoid confounders due to the type and duration of treatment. We did not identify an association ($p = 0.051$, Supplementary Figure 8A) between patients with amplicons, which includes BFBs and ecDNAs, and a poorer survival that was reported in previous studies¹⁻³.

2. Long-read methylation calls were used to disentangle amplicon assembly graphs. Did the authors observe that most of the amplicons classified as ecDNA were unmethylated? Figure 2E suggests that this may not always be the case.

We identified that the methylation patterns within ecDNA in our study vary between tumours. There were two tumour showing a higher variation in methylation patterns (018T and 139T) in ecDNA regions, whereas three tumours show lower variation in regions sampled including those identified to be part of the ecDNA circles (Supplementary Figure 8B-C).

To explain the variation between tumours, we determined the methylation status of the a set of sequences in all regions found in ecDNA circles across the long read genomes. This set of genomic regions allows for the comparison of methylation differences between tumours and ecDNA regions. The set of regions contain the same sequence, number of promoters and enhancers elements to discern the differences in methylation due to the presence of the ecDNA circles.

To further investigate the differences in methylation in tumours, we identified differential methylation in regions annotated by the epigenome road map and identified regions in enhancers (H3K27Ac, H3K4Me1) and heterochromatin marks (H3K27me3, H3K36me3 and H3K9Me3) that were differentially methylated (wilcox test FDR < 0.1) in individual samples (Supplementary Figure 8B). We speculate that the amount of differential methylation in each ecDNA is determined by overall methylation levels in a tumour genome and the presence of large regions in the ecDNA containing either enhancers or heterochromatic marks.

We have added the below text to line 412 in the discussion section and the figures in supplementary figure 8A-B.

We identified that the methylation patterns within ecDNA in our study varied between tumours. In addition, we compared the fraction of methylated reads between samples and regions with and without ecDNA amplicons in each tumour. Despite a small set of tumours, we observed that the methylation patterns are determined by the overall methylation levels in a tumour genome and the presence of large regions containing enhancers or heterochromatic marks (Supplementary Figure 8B-C).

Supp Figure 8B-C . Methylation status of genomic DNA and ecDNA sequences. B) Boxplot showing fraction of methylation in each tumour, in a union set of genomic regions present in the long read genomes and if regions are part of an ecDNA amplicon in each tumour. C) Boxplot showing fraction of methylation of each tumour split by epigenome roadmap annotations (H3K27Ac, H3K4Me1, H3K27me3, H3K36me3 and H3K9Me3). Wilcoxon rank sum test was used to identify differences in fractions of methylation between regions in ecDNA and not in ecDNA in a tumour.

3. In the cases with decreased BFB CN prevalence in organoids, it would be interesting to see if this was a result of aneuploidy, since BFB cycles may lead to chromothripsis or LOH.

We re-analysed the organoid with decreased BFB CN prevalence and found that there is no evidence of aneuploidy such as chromothripsis or LOH. We used Hatchet⁴ to model allele-specific CN profiles and found that the BFB found in the tumour containing CDK6 (black arrow, chr7) is only present in the tumour and not in the clone expanded in the organoid. The CN profile in both the tumour and organoid do not show any evidence of chromothriptic events (oscillating CN profiles) and no evidence of LOH.

This recapitulates our findings that the BFB events can be used to track clones since they can be inherited stably in clones that expanded in the organoid.

Copy number profile of organoid CAM412 and matched tumour. Copy number profile in tumour showing an amplified segment containing a BFB (black arrow) that is absent in the organoid, with no evidence of additional copy number alterations in the region.

4. Supplementary table 1 provides a summary-level information of the cohort. Is it possible to add information about individual patients, e.g. amplification status, treatment, smoking status, race, etc? In addition, it would be more convenient if the tables were in excel format, rather than pdf.

We thank the reviewer for the comment and have added supplementary table 2 to include patient level information with amplification statuses, treatment and other risk factors including smoking, alcohol use and history of reflux.

5. The relevant custom code / pipelines should be made available via a public software repository

We have included the custom code for ecDNA assembly in the github repository and is accessible from <https://github.com/fitzgerald-lab/ecAssemble>.

Minor:

1. In the summary figure, it would be informative to add the number of samples that were subjected to each sequencing method.

We have added the number of the samples subjected to each sequencing method in the summary figure. In summary, we carried out short read sequencing to 710 tumours and 24 organoid models, long read sequencing on 9 tumours, 3 normal tissue and 3 organoids and scDNA sequencing at 2 passages for CAM277.

Reviewer #2 (Remarks to the Author):

“I co-reviewed this manuscript with one of the reviewers who provided the listed reports. This is part of the Nature Communications initiative to facilitate training in peer review and to provide appropriate recognition for Early Career Researchers who co-review manuscripts.”

Reviewer #3, expertise in esophageal cancer genomics (Remarks to the Author):

The manuscript by Alvin et al. reported important insights into the Complex amplifications of EAC based on WGS and long-read sequencing, especially for ecDNA and BFB. The WGS of 710 patients comprehensively characterized the high copy number amplification mechanisms of the EAC genome. In addition, the author conducted long-read sequencing on 9 EAC patients and used de-novo assemblies and DNA methylation to reconstruct complex ecDNA and BFB amplicons, effectively solving the problem of constructing complex chromosome rearrangements in the short-read sequencing. Finally, the author integrated the patient-derived organoids and single-cell DNA sequencing to explore the ecDNA clonal dynamics of EAC tumor evolution.

In summary, they present a series of bioinformatic works to decode the characteristics of these events using methylation, long sequencing reads, as well as single-cell sequencing data. They found that complex amplicons are already present in the early stage, and confirmed the genomic consistency between tumor and patient-derived organoids. They also illustrate the importance of using long-read and single-cell DNA sequencing to track EAC evolution. Although some of the conclusions are expected, they did provide insights into the complex rearrangement of different perspectives. In addition, some concerns need to be addressed:

Major comments:

1. The author analyzed the high copy number amplification mechanisms in EAC, including ecDNA, BFB, Complex non-cyclic and Linear amplification. In previous studies on structural variations, it has been mentioned that complex tandem repeats can lead to high copy number amplification (PMID: 32025012, 33007263, 36272974). Therefore, can the author refine the classification of this part and analyze their impact on oncogenes? Because complex tandem duplications play an important role in the high copy number amplification and regulating the expression of driver genes.

We thank the reviewer from the insightful comments. We agree that complex tandem duplications are likely to have an effect in driving copy number amplifications and oncogene amplification and refined the categories of amplicons. However, there are substantial overlap in SV categories outlined in Li et al 2019⁵, between the *cycles* and ecDNA events identified in this study. It is unclear if they should be distinct categories, hence we opted to keep the BFB and ecDNA categories as they are easier to interpret. As prompted by the reviewer, we refined the category of complex non-cyclic events by analysing tandem duplications and their functional effects, i.e resulting in a duplication of an oncogene.

In the complex non-cyclic events, we identified a lower average amplified copy number (CN = 6.3 (IQR 5.2-7.7) compared to BFBs and ecDNA (CN = 7.8 (6.0-12.8), CN = 12.6 (6.1-8.3)). In these complex non-cyclic events, 176 (67%) had a oncogene identified in the amplified region. We next analyzed the types of SV events in these regions and identified that most clusters consist of a mixture of different SV types (INS, INV, DEL, DUP, BND) and used the same schema as Li et al. to categories the events. We split the events according to whether the duplication overlapped with regions of the identified oncogene and found 123 (47%) to result in a duplication in an oncogene. Lastly, we estimated the percentage of duplications as a metric to identify complex clusters with a large number of duplications similar to the pygro events identified in Hadi et al⁶.

There was a wide variation in the percentage of duplications in each complex cluster in both events, with and without affecting oncogenes, and remains a challenge to systematically categorize these complex events.

Characterization of complex tandem duplication (TD) containing clusters.

A) Boxplots showing average amplified copy number for BFB, complex non-cyclic and ecDNA events. B) Boxplots showing average copy number for complex TD events with oncogene present in the amplified interval. C) Percentage of duplication events in the cluster in groups with oncogene present and not present in the amplified intervals.

2. The author assembled high copy number amplicons of 9 EAC patients through long-read sequencing. The manuscript mentioned the high concordance in the results of ecDNA between AA and de-novo assembly. However, the author did not provide clear results of other complex amplification events. Is there any difference in these results? If there are differences, what are the reasons for them?

We thank the reviewer for the comments. We did not comment on the complex amplification events as AA is designed to call ecDNA and BFB events in amplified regions (> 4.5 copies) of the genome. The complex amplification events are hard to resolve and compare since they might arise from different mechanisms and AA combines these events together into the complex non-cyclic category if they do not fall into categories of linear amplification, BFB and ecDNA. In addition, the complex non-cyclic events are challenging to classify and compare, as shown in the response for reviewer 3 comment 1.

3. When analyzing the amplification of 9 long-read sequencing data, the author only used 3 paired normal genomes as controls. How were these three EAC patients selected? Why didn't you do it all? More details are needed to introduce the matched normal samples. The abundance of a mobile element or L1 sequence near ecDNA might be germline events.

We thank the reviewer for the comments. As we have already sequenced the normal genome of each case on the Illumina platform, we selected these three EAC normal samples to sequence, to assess the benefit of sequencing both tumour and normal tissues using long reads. We reached the conclusion that a second matched normal

sample sequenced using long reads did not provide much benefits as we could already get SNV and LINE1 insertions based on the Illumina sequenced normal.

We carried out additional analysis on the L1 sequences near ecDNA and identified that the L1 insertions are somatic and not found in the normal tissues. This analysis is shown in Supplementary figure 4A & 4B where we showed the IGV screenshots of the L1 insertions in the tumour and but not in the paired normal sample.

4. For derived organoids, the manuscript mentions that the amplicon events are highly concordant between paired tumor and organoid (Line 264-266). Previous study has shown that some EAC patients have intra-tumoral heterogeneity (PMID: 31949146). How can the author avoid the troubles caused by intra-tumoral heterogeneity? The results show that there seems to be no impact of intra-tumoral heterogeneity, Are there specific requirements when selecting tumors? Among them, an organoid (CAM277) with an amplicon CN profile that differed from the primary tumor, and may heterogeneity also be one of the reasons?

In the OCCAMS study, we set a purity requirement of 0.7 for a tumour to be sequenced according to International Cancer Genome Consortium (ICGC) standards. This mitigates low sample purity and intra-tumour heterogeneity to an extent. The differences in amplification profile in CAM277 is indeed due to heterogeneity, or that multiple distinct clones have been sampled during organoid derivation and expanded across time. To mitigate this it is possible to derive multiple organoids from single cells per patient – however this is outside of the scope of this project.

5. The findings tend to be descriptive. Many observations are obtained through sporadic samples with long-read sequencing, but it seems difficult to get a solid conclusion. As 710 tumors are sequenced, is there any way to further validate the presence of mobile element insertion as a driver of complex amplicons' origin?

In an ideal world one would increase the sample size for long read sequencing but it is a formidable and expensive exercise and the purpose of this study is to show the potential and added value from long read sequencing. We and others⁷ have observed cases where L1 insertion events are associated with complex amplicons. Despite limitations in short read sequencing to resolve L1 insertions, we identified 43 tumours with L1 insertion events (6% in the 710 tumours that show that it is a plausible mechanism that is associated with the origin of complex amplicons.

These L1 events were found <50Kb of amplicon events, that can be associated with the amplicons, but there is a dearth of methods to definitively show the role of L1 in the formation especially in tumours with highly complex amplicons. We have shown the 139_T case shown in the main figure compared with three examples of additional events discovered in Supplementary Figure 4C-E.

The following text has been added to the main text in line 240 :

To identify tumours in the 710 cohort with LINE-1 insertions near complex amplicons, we integrated TraFic⁷ MEI and AA calls and identified an additional 43 tumours. Despite limitations in short read sequencing to resolve L1 insertions, we show that it is a plausible mechanism that is associated with the origin of complex amplicons.

Supplementary Figure 4C-E. Examples of L1 insertions within 50kB of complex amplicon events.

Minor

comments:

1. Line 79, Figure 1 is not well-organized, as it seems that there are two Figure 1 in the manuscript.

We apologise for the confusion. The first figure in the manuscript was meant to be an summary figure and we have mistakenly numbered the figure 1 as well. We have changed the figure number in the current version.

2. Line 101, did the author notice that many amplicons contain the enhancer of oncogenes only, such as gene MYC and KLF5. Thus, these amplicons should be considered if omitted.

Indeed we identified amplicons containing enhancers of oncogenes and in our analysis, we included those events as we observed that they can be present in amplicons with

possible interaction with additional oncogenes on other chromosomes. These events are a minority compared to amplicons that contain oncogenes and we believe these events can be analyzed in further studies to study the yet unknown oncogenes or unknown roles these amplicons play in cancer.

3. Line 167 Lack of legend for A Graph

We have added a legend for Figure 2A.

4. Line 181. Again, are there any matched normal samples subjected to long-read sequencing for these three patients?

We have sequenced these cases on short read platforms and hence sequencing these matched normal on long reads did not yield much additional benefit.

5. As the accuracy and sensitivity of SVs are the foundation of this study, a detailed method for SV calling for both short- and long-reads should be shown.

We have added an additional section in the methods with the below text on line 457:

Structural variants were called using Manta⁸ as previously reported⁹, for the 710 short read tumours. In addition, we carried out integrated CNV and SV calling using the GRIDSS-Purple-LINX¹⁰ suite using default parameters, to allow for the comparison of SVs in the tumour and organoids. LINX annotation of complex clusters was used to further annotate complex non-cyclic events to identify the pattern SV types in each cluster.

6. Line 230 Figure 3J annotated error

We have amended the error to reflect figure 2J.

7. Please label chromosome numbers in Figure 3B-F

We have added the chromosome labels in 3B-F

8. Please describe the legend of Figure 3H-I in more detail.

We have added the below text into the legend of figure 3H-I

H) Interphase FISH of CAM277 showing KRAS ecDNA (red) against centromere labeling (green) for chromosome 12 (CEN12q). I) Metaphase FISH of organoid CAM277 with additional DAPI staining for DNA, red for KRAS and green for CEN12q. Yellow arrows denote red fluorescence signal of ecDNA outside of chromosomal structures. 40X magnification used for interphase and metaphase FISH.

9. It is best to draw the copy number of ERBB2 in Supplementary Figure 1

We have added the ERBB2 copy number of each tumour. The range of CN values are CN 2-5.

10. Please consider labeling the H3K27Ac signal and the H3K27Me3 regions in the DMR region(Figure 2E)

We have added a full version of Figure 2E with labels of the H3K27Ac and H3K27Me3 signals in supplementary figure 3A.

11. Line 277 GRIDSS-LINX needs to annotate literature sources. In addition, please add the description of GRIDSS-LINX to the method.

We have added GRIDSS-Purple-LINX to the methods and citations

12. Line 529, the title of the section 'Availability of data and materials' should be bold.

We have fixed the title and bold the text.

Reviewer #4, expertise in single cell genomics and esophageal cancer (Remarks to the Author):

In this manuscript, the authors explore the genomic distribution and structure of gene amplification structures as extrachromosomal DNA and BFBs in esophageal adenocarcinoma. Overall, this study exhibits a robust foundation, as the results are substantiated by a diverse range of experimental and analytical data. Notably, the application of Nanopore sequencing, patient-derived organoids and single cell analysis for investigating extrachromosomal DNA and BFBs yields intriguing findings. The use of Nanopore sequencing to validate ecDNAs and BFBs and the discovery of potential associations with hypomethylation states are significant findings. Furthermore, their exploration of these amplicons in tumor organoids and the tracking of their dynamics in serial passages contribute to uncovering their origins and functions. Collectively, these findings position this manuscript as a relatively novel and comprehensive study in the field. While the data processing is generally robust, it would be beneficial to provide more detailed information in the methods section and supplementary tables.

However, in several instances, the manuscript appears somewhat unrefined and requires further polishing.

1. The analysis initially focuses on a '710 Cohort.' However, there appears to be no description provided for this dataset. It's important to clarify whether this omission is deliberate for the purpose of this study or if the same dataset has been utilized in previous projects. Exploring the associations between extrachromosomal DNA (ecDNA) and their clinical implications, as well as examining SNPs, holds valuable insights that could benefit the broader scientific community.

We thank the reviewer for the comments. We have described the 710 cohort in a previous publication showing the mutational processes in EAC¹¹. The same dataset was used to build on the previous analyses, to add the role of complex amplicons in describing the clinical correlations in EAC. We agree with the reviewer that the associations with ecDNA are valuable to the broader community and we did not identify an association of the presence of amplicons with poorer survival ($p < 0.051$) in the 710 cohort (Supplementary Figure 8A).

Supplementary Figure 8A. Kaplan Meier plots of patients in the 710 with an amplicon (BFB or ecDNA) compared with patients without amplicon events.

2. In Figure 1A, it appears that KRAS has a peak dominated by BFB, which does not align with the observations presented in Figure 1C.

We thank the reviewer for the comment and clarify that the plot shows the cumulative frequencies of the regions amplified through BFBs as a mechanism. Figure 1C provide additional details on individual segments amplified (sorted by start position) and the visualization of the bottom 6 segments differ from the cumulative plot.

3. As emphasized by the authors, it is noteworthy that BFB, ecDNA, and CNCs exhibit biases in genome regions. Therefore, it is advisable to include discussions and results regarding their distinctions in cellular functions and potential clinical implications.

We thank the reviewer for the insightful comments and have added additional discussion points regarding the distinction in cellular functions shown below for convenience.

The identification of initiating rearrangements and copy number changes in known amplicon regions may provide a useful biomarker for the early detection of EAC in the clinical setting. Click or tap here to enter text. *The differences and biases in genomic regions in the initiation of BFBs and ecDNA can be due to that sequence of the region or the presence of regulatory elements. Following the initiation event, CN gain leads*

to downstream effects including over-expression or gene regulation of nearby genes. Over-expression of oncogenes, based on the number copies of a gene, is limited to the number of BFB cycles that occurred whereas the formation of ecDNA can bypass this limitation due to the stochastic inheritance of oncogenes in subclones. We have shown that the presence of ecDNA can be observed early in dysplastic BE¹² and in our study with P43 with the same ERBB2 ecDNA in both the Barrett's and tumor biopsy. It may be possible to risk stratify BE patients according to evidence of any initiating events or amplified regions. We expect these amplicon events to be at a lower CN compared to the tumors and obscured due to the presence of multiple subclones in an earlier stage of disease. Therefore, developing approaches to detect these events early in the pathogenesis of cancer is an area for further research.

4. In Figure 2B, it is observed that these two samples exhibit a relative copy number of 2-3, which does not appear to be indicative of extrachromosomal DNA (ecDNA). Furthermore, their methylation patterns closely resemble those of normal samples.

We thank the reviewer for the comment. We apologize for the issues in visualizing the regions which is because there is a lack of tools that can call copy number for a small region reliably, especially whilst modelling events such as WGD in the tumour. We have re-called the CNVs for this pair of samples using HATCHET2.0⁴ and estimated the CN of the ERBB2 event in the tumour to be 41 copies with evidence of WGD. The CN for the BE case on the other hand was harder to be estimated due to the small region, at about CN 7-10.

In addition, we have re-analysed the methylation patterns of the normal, BE and tumour sample once more and identified shared regions that are demethylated in the normal, BE and tumour that was not identified in the panel of normal samples (Supplementary Figure 3A).

5. As in the BE (normal) sample, in the majority of cases, their copy numbers remain within the normal range. It is indeed surprising to detect a copy number (CN) and even an extrachromosomal DNA (ecDNA) with a copy number of 7.

We understand that this might seem surprising. We and others have shown the CN profiles of BE cases have a variety of CN profiles^{13,14}, and shown that BE cases that progress to cancer have a larger number of CNVs. We have shown the presence of ecDNA at pre-cancer stages of disease¹² and this observation supports the findings.

6. Numerous graphs depicting ecDNA and BFBs, such as those found in Figure 2 and others, raise the question of whether these representations reflect the actual complexity of DNA structures that form and replicate within cells, or if they are rough assemblies of individual amplicon structures?

We have developed the assembly graphs as robust representations of the amplicons by using only the regions amplified in the ecDNA for the construction of these graphs. This approach refines the graph by removing non-amplified chromosomal segments and reduces sequences that are at a low copy number. We ensured the sequence quality of the assemblies and ensured that the graphs are representative of the amplicons.

To test if the graphs are representative of the amplicons that form, replicate and get passed down in cells, we refined the assembly graph of CAM277 using the hypomethylated reads. The methylation refined graph recapitulated the sequences with the highest correlation with edge_25 (containing *KRAS*) shown in the heatmap in figure 4E.

Supplementary Figure 3H. Methylation refined assembly of CAM277 recapitulated edges with highest correlation with *KRAS* containing sequences shown in figure 4E

7. The second section of the results appears to be quite lengthy. It might be beneficial to consider dividing it into two separate sections for improved organization.

We thank the reviewer for the comments and reorganised the section into two separate sections:

- 1) *De-novo* assembly of long reads classifies complex amplicons into BFB and ecDNA events and
- 2) Long read assemblies resolve complex amplicons and identify initiating processes.

8. The association between the hypomethylation state and ecDNA reads appears to be underaddressed. Is this conclusion widely applicable and firmly established, or does it apply only to specific samples? The graph in Figure 2J presents a promising analysis; it might be worthwhile to conduct similar analyses for other samples as well.

We thank the reviewer for the comment that was also brought up by reviewer 1. We identified that the methylation patterns of ecDNA in our study vary between tumours,

with two tumours showing higher variation in methylation patterns (018T and 139T) in ecDNA regions whereas three tumours show lower variation in regions sampled in the genome and regions identified to be part of the ecDNA circles (Supp Fig 8B-C). To explain the variation between tumours, we determined the methylation status of the sequences in all regions found in ecDNA circles in the long read genomes. This set of genomic regions allows for the comparison of methylation differences between tumours and ecDNA regions. The set of regions contain the same sequence, number of promoters and enhancers elements to discern the differences in methylation due to the presence of the ecDNA circles.

In addition we re-analysed the ecDNA circles as recommended and found that 3 samples with ecDNA had focal demethylation patterns and benefited less from using the methylation patterns for refining the assemblies We identified CAM277 and 65T to have cyclic amplicons after refining the amplicons using the hypomethylated read for assembly.

Figure 3G-H: Cyclic assembly graphs refined using hypomethylated read for assembly.

9. In Figure 3A, the MYC seem to be detected also in tumor.

We have compared the ecDNA calls based on Amplicon Architect calls and plotted the CN profiles for visualization. We agree that some breaks in the CN profile hints that the MYC amplicon might have already been present in the tumour but a CN of 3 prevented the detection of the event by AA due to the low CN.

10. In Figure 3C, it appears that the organoids may have lost some of the amplicon details. Could this be attributed to changes in the amplicon structure within such a small genome region?

We thank the reviewer for the insightful comments. We re-analysed CAM453 using GRIDSS-LINX calls and identified a cluster of inversions in tumour that was absent in the organoid. Both the tumour and organoid shared a duplication event (chr8:126694685-130657526) that increased from 0.4-108 copies. We speculate that the cluster of inversions was independent of the amplicon structures, and were lost when a clone with the MYC amplicon expanded and other clones with SVs within that genomic region were lost.

Table 1:SV events lost between tumour and organoid in CAM453

ClusterId	Type	ChrStart	PosStart	PosEnd	JunctionCopyNumber
109	INV	chr8	127370698	127373469	1.5695
109	INV	chr8	127370723	127373517	0.3439
109	SGL	chr8	127373668	0	0.5807
109	INV	chr8	127450418	127605858	0.7411
109	INV	chr8	129217439	129217629	0.7813
109	DUP	chr8	126694685	130657526	0.4084
129	SGL	chr8	144263536	0	1.2809
129	BND	chr8	144263540	50462400	0.9038

11. In Figure 3E, it's noteworthy that the tumor exhibits a higher copy number (CN) than the pure organoid, and this requires further explanation. Additionally, the sudden reappearance of the initial ecDNA structure in P14 appears suspicious. It is more likely that P0 and P8 did not fully detect the amplicon in detail and may have missed them.

We agree and it was this observation that prompted us to carry out single cell sequencing of that organoid and showed further evidence of a clonal switch between passages in Figure 4.

12. In Figure 4F, the detection of the loss of a BFB amplicon is noteworthy. Given the availability of results from patient-derived organoids (PDOs) across multiple passages, it would be beneficial to conduct additional analyses to explore the evolutionary patterns of various types of amplicons.

We thank the reviewer for the comments and we have carried out an additional analysis to identify all of the changes of amplicons between passages. We used both the scDNA copy number profile to identify bins with high variance between passages and used destruct (<https://github.com/amcpherson/destruct>) to genotype SVs based on the raw single cell reads. Of note, we identified an additional complex event on chromosome 4 & 7 in clone C which was depleted upon passaging, present in P4 but not P15. Conversely a complex cluster on chr12 containing C12orf77 and KRAS was found to be enriched in P15 in both clones C and D.

Supplementary Figure 7. CNV and SV events different between passages after clonal shift in CAM277. A) Histogram showing CN values of cells at P4 and P15 in a region with a complex amplicon on chr4 and B) Heatmap of SV events of the chr4 complex amplicon, C) Histogram of a genomic bin on chr12 showing a higher CN at P15 and D) SV events on chr12 enriched at P15

13. Line 230, is it Supplement Figure 3J?
 We apologise for the error as it is indeed Supplement Figure 3J.

Reviewer #5, expertise in esophageal cancer organoids (Remarks to the Author):

In this manuscript, Ng, Fitzgerald, and colleagues present a study on oncogenic amplicons in oesophageal adenocarcinoma. They re-examine OCCAMS WGS data using AmpliconArchitect to describe the landscape of amplicons in OAC. They then use long read assemblies to study more complex amplicon structures in a subset of samples. Finally, they use organoids derived from OAC tissue to study the evolution of oncogenic amplicons during organoid culture and passage. While largely observational, this study uses cutting edge techniques to study the impact of amplicons on OAC tumourigenesis and act as a proof-of-concept for further study.

Major comments:

1. How were those 24 organoids chosen?

We thank the reviewer for the comment. The organoids with whole genome sequencing was used for this study that showed robust growth after passaging were all included for this study.

We have added a sentence in the method section to describe how the organoids was chosen in line 510.

2. What happened to the data from the 7 organoids where you did not find amplicons with recurrently altered OAC oncogenes? For those 7 organoids, did the primary tissue have similar amplicons or did it have amplicons with recurrently altered OAC oncogenes? Did these 7 organoids recapitulate the primary tissue otherwise? Along those lines, does this imply that 30% (7/24) of organoids/tumours do not have amplicons as a mechanism of tumour propagation?

We thank the reviewer for the comments. In these 7 organoids, we did not find any amplicons in recurrently amplified genes and we speculate that there are other mechanisms of tumour development. In a previous study of 383 tumours⁹, we identified subtypes of tumours based on the rearrangement profiles and the presence of amplicons is one of the mechanisms for tumour development. In addition, there are tumours driven by deletions, L1 insertions, tandem duplications and tumours with a lack of rearrangements.

These tumours recapitulate the primary tissue based on SNV and CNV profiles despite the lack of an amplicon in a driver gene.

3. Depending on above answer, the 94% concordance reported does not take into account the 7 removed organoids.

That is correct as we have determined the concordance calculation based on whether there is a detectable amplicon in the tumour and the ground truth based on the presence of an amplicon.

We added a line in the results line 265 with the following clarifications:

We did not find any amplicons in 7 organoids and their corresponding tumours and omitted them for further comparisons.

4. For Figure 3A, what passage were the organoids taken to compare to the tumour?

We thank the reviewer for the comment. We have included the passage information for the organoids in Figure 3A in supplementary table 6.

5. Given stochastic inheritance and the multiple clones demonstrated by your scDNAseq, how do you think the heterogeneity of the organoids in culture at a particular passage affects the interpretation of the organoid data? Could the detection of ecDNA poorly represented in primary tissue be a result of de novo clones?

We thank the reviewer for the insightful comments. We think that the heterogeneity of the organoids is an important factor in the interpretation of organoid data especially in the clinical settings. It is of utmost importance to ensure that the derived organoid recapitulate the tumour, regardless of the presence of a single or multiple clones in a tumour. We believe that carrying out whole genome sequencing of the primary tissue and derived organoid at P0 is an important technical step to ensure that the organoid resembles the primary tumour. Furthermore, additional sequencing has to be done prior to any clinical applications such as drug testing or drug screening assays to ensure there are no clonal shifts during passage.

The question of whether the ecDNA events were present in tumour or due to de-novo clones is an important point and we have addressed the points by comparing the breakpoints of the rearrangements as it is unlikely to have two separate SV events at the same genomic position in two independent clones. We have used the GRIDSS SV calls to verify that several breakpoints were detectable at a low CN in the tumour and is due to clonal expansion after the organoid was derived and not a de-novo clone.

6. The authors demonstrate that organoids will clones harboring ecDNA expand during culture due to a competitive advantage. The authors suggest that this may allow for the modelling of clone fitness in vitro. I think that a potential downside is that the in vitro environment poses a different selective environment than in vivo and may not serve as an accurate representation of the in vivo evolution. This might be interesting in the discussion.

We fully acknowledge that the downside of the *in-vitro* organoid model will post a different selective environment than *in-vivo*, especially without the constraints in the tumour microenvironment and other immune cell types.

On the other hand we believe that the application of using tumour organoids will provide a clean experimental model to determine clonal fitness based on the genetics alone and provide insight into the proliferative advantages of clones based entirely on the basis of genetic alterations. We foresee clones that benefit from other selective adaptations to hypoxia or immune evasion to not be modelled well in the organoid system.

We have added a comment to this effect in the discussion at line 436

The limitation of this model is that the in-vitro organoid model will post a different selective environment than in-vivo, especially without the constraints in the tumour microenvironment and other immune cell types.

Minor

comments:

1. How did you use the 3 paired normal genomes for Nanopore?

We thank the reviewer for the comment. As we have already sequenced the normal sample of each case on Illumina platform, we selected these three EAC normal samples to assess the benefit of sequencing both tumour and normal tissues using long reads. We reached the conclusion that a second matched normal sample sequenced using long reads did not provide much benefits as we could already call SNV and L1 insertions based on the Illumina sequenced normal. As such we used them to compare methylation profile of the same tissue type where a paired normal was not sequenced on long reads.

2. Some detail on how the organoids were harvested for WGS, depth of sequencing, etc, in the methods would be helpful.

We thank the reviewer for the comment. As such we have included information on organoid derivation, depth of sequencing and other details in the methods section at line 506.

Briefly, organoids were derived by collecting tissue from surgical resection, of which half of the tissue were used and the remaining snap frozen for genomic profiling. The snap frozen tissue was stained with Haematoxylin and Eosin and the cellularity of the sample was reviewed by two pathologists independently. Tissues with $\geq 70\%$ cellularity underwent DNA and RNA extraction using the AllPrep Kit (Qigen) and sequenced on paired-end Illumina sequencing to a depth of 30x. Blood or normal squamous esophageal samples were selected as germline reference samples.

References

1. Kim, H. *et al.* Extrachromosomal DNA is associated with oncogene amplification and poor outcome across multiple cancers. *Nat Genet* **52**, 891–897 (2020).
2. Decarvalho, A. C. *et al.* Discordant inheritance of chromosomal and extrachromosomal DNA elements contributes to dynamic disease evolution in glioblastoma. *Nat Genet* **50**, 708–717 (2018).
3. Wu, S. *et al.* Circular ecDNA promotes accessible chromatin and high oncogene expression. *Nature* **2019 575:7784** **575**, 699–703 (2019).
4. Zaccaria, S. & Raphael, B. J. Accurate quantification of copy-number aberrations and whole-genome duplications in multi-sample tumor sequencing data. *Nat Commun* **11**, 4301 (2020).
5. Li, Y. *et al.* Patterns of somatic structural variation in human cancer genomes. *Nature* (2020) doi:10.1038/s41586-019-1913-9.
6. Hadi, K. *et al.* Distinct Classes of Complex Structural Variation Uncovered across Thousands of Cancer Genome Graphs. *Cell* **183**, 197-210.e32 (2020).
7. Rodriguez-Martin, B. *et al.* Pan-cancer analysis of whole genomes identifies driver rearrangements promoted by LINE-1 retrotransposition. *Nat Genet* **52**, 306–319 (2020).

8. Chen, X. *et al.* Manta: Rapid detection of structural variants and indels for germline and cancer sequencing applications. *Bioinformatics* **32**, 1220–1222 (2016).
9. Ng, A. W. T. *et al.* Rearrangement processes and structural variations show evidence of selection in oesophageal adenocarcinomas. *Commun Biol* **5**, 1–12 (2022).
10. Shale, C. *et al.* Unscrambling cancer genomes via integrated analysis of structural variation and copy number. *Cell Genomics* **2**, 100112 (2022).
11. Abbas, S. *et al.* Mutational signature dynamics shaping the evolution of oesophageal adenocarcinoma. *Nat Commun* **14**, 4239 (2023).
12. Luebeck, J. *et al.* Extrachromosomal DNA in the cancerous transformation of Barrett's oesophagus. *Nature* (2023) doi:10.1038/s41586-023-05937-5.
13. Katz-Summercorn, A. C. *et al.* Multi-omic cross-sectional cohort study of pre-malignant Barrett's esophagus reveals early structural variation and retrotransposon activity. *Nat Commun* **13**, (2022).
14. Killcoyne, S. *et al.* Genomic copy number predicts esophageal cancer years before transformation. *Nat Med* **26**, 1726–1732 (2020).

Reviewers' Comments:

Reviewer #1:

Remarks to the Author:

I thank the authors for addressing all my requests in detail. I have no further comments.

Reviewer #2:

Remarks to the Author:

Reviewer #3:

Remarks to the Author:

For the first major comment, the author added an analysis of the categories of complex non-cyclic events and the frequency of their impact on oncogenes. I would like to ask if there are hot spots in these events or regions that can serve as potential therapeutic targets? If such events exist, it is recommended that the author list them or consider further discussion in future research.

I don't have any other questions, and the author has basically answered my doubts.

Reviewer #4:

Remarks to the Author:

The author has satisfactorily addressed all pertinent queries. In my opinion, it is fit for publication.

Reviewer #5:

Remarks to the Author:

The authors have adequately addressed my questions. Thank you and congratulations on a nice paper.

Reviewer #1 (Remarks to the Author):

I thank the authors for addressing all my requests in detail. I have no further comments.

We thank the reviewer for reviewing this paper.

Reviewer #2 (Remarks to the Author):

We thank the reviewer for reviewing this paper.

Reviewer #3 (Remarks to the Author):

For the first major comment, the author added an analysis of the categories of complex non-cyclic events and the frequency of their impact on oncogenes. I would like to ask if there are hot spots in these events or regions that can serve as potential therapeutic targets? If such events exist, it is recommended that the author list them or consider further discussion in future research.

I don't have any other questions, and the author has basically answered my doubts.

We thank the reviewer for the question. We have listed the hot spots affected by complex non-cyclic events in line 336. We do not understand and appreciate the role of complex non-cyclic events in the disease and will consider discussion in future research.

We detected hotspots with complex non-cyclic events affecting GATA6, GATA4 and CCND1 that can have an under-appreciated effect on EAC pathogenesis.

Reviewer #3 (Remarks on code availability):

The code provide a README file with enough instructions for installing and running the application. I can install and use the software normally, but I have not tested the accuracy of the results after running the software.

We thank the reviewer for testing the software.

Reviewer #4 (Remarks to the Author):

The author has satisfactorily addressed all pertinent queries. In my opinion, it is fit for publication.

We thank the reviewer for reviewing this paper.

Reviewer #5 (Remarks to the Author):

The authors have adequately addressed my questions. Thank you and congratulations on a nice paper.

We thank the reviewer for reviewing this paper and the encouraging comments.